# A proteomics landscape of circadian clock in mouse liver

Yunzhi Wang[1], Lei Song[2], Mingwei Liu[3], Rui Ge[1], Quan Zhou[3], Wanlin Liu[3], Ruiyang Li[1], Jingbo Qie[1], Bei Zhen[3], Yi Wang[3,4], Fuchu He[1,2,3], Jun Qin[1,3,4] & Chen Ding[1]

As a circadian organ, liver executes diverse functions in different phase of the circadian clock. This process is believed to be driven by a transcription program. Here, we present a transcription factor (TF) DNA-binding activity-centered multi-dimensional proteomics landscape of the mouse liver, which includes DNA-binding profiles of different TFs, phosphorylation, and ubiquitylation patterns, the nuclear sub-proteome, the whole proteome as well as the transcriptome, to portray the hierarchical circadian clock network of this tissue. The TF DNA-binding activity indicates diurnal oscillation in four major pathways, namely the immune response, glucose metabolism, fatty acid metabolism, and the cell cycle. We also isolate the mouse liver Kupffer cells and measure their proteomes during the circadian cycle to reveal a cell-type resolved circadian clock. These comprehensive data sets provide a rich data resource for the understanding of mouse hepatic physiology around the circadian clock.

[1] State Key Laboratory of Genetic Engineering, Human Phenome Institute, Institutes of Biomedical Sciences, School of Life Sciences, Zhongshan Hospital, Fudan University, Shanghai 200032, China. [2] School of Life Sciences, Tsinghua University, Beijing 100084, China. [3] State Key Laboratory of Proteomics, Beijing Proteome Research Center, National Center for Protein Sciences, Beijing 102206, China. [4] Alkek Center for Molecular Discovery, Verna and Marrs McLean Department of Biochemistry and Molecular Biology, Department of Molecular and Cellular Biology, Baylor College of Medicine, Houston, TX 77030, USA. These authors contributed equally to this work: Yunzhi Wang, Lei Song. Correspondence and requests for materials should be addressed to F.H. (email: hefc@nic.bmi.ac.cn) or to J.Q. (email: jqin@bcm.edu) or to C.D. (email: chend@fudan.edu.cn)

The mammalian circadian clock includes a "master clock" within the suprachiasmatic nucleus (SCN) of hypothalamus and "peripheral clocks" within other tissues of the body. The "master clock" functions as an "orchestra conductor" to direct "peripheral clocks" through yet-to-be defined pathways[1], allowing animals to adapt their feeding, activity, and metabolism to predictable daily changes in the environment.

Circadian clocks orchestrate physiological rhythms via the temporal regulation of gene expression to control core clock genes and rhythmic output programs. A network of transcriptional–translational feedback loop comprised of core transcriptional activators (Bmal1 and Clock) and repressors (Per and Cry), to control the rhythmicity in gene expression[2,3]. Tp53 and Myc, which are well-characterized cancer driver genes, and several multi-functional nuclear receptors (NRs) including Reverb, Ror, and Ppar family[4–6], have also been shown as important regulators of the circadian clock. These studies demonstrate the critical roles of TFs in regulating circadian rhythm.

Liver plays a fundamental role in circadian clock system. Transcriptome profiling of the liver has demonstrated the circadian variation in the expression of genes related to oxidative metabolism, mitochondrial functions, and amino acid turnover[7], and that transcriptional regulation drives the circadian mRNA rhythms[8]. In contrast, much less is known on the protein level. With rapid development of analytical techniques[9], particularly mass spectrometry-based proteomics, it is increasingly feasible to measure proteins in order to understand the diverse biological processes. Recently, Robles et al.[10] and Wang et al.[11] reported proteome studies in circadian clock of the mouse liver[10,11]. However, due to the technical limitations in proteomics techniques applied, the dynamics of transcription factors—the key drivers of gene regulations around the circadian clock, were still poorly understood.

It is expected that a hierarchical circadian regulation network might exist, which may include different regulatory layers that facilitate signal transductions around the clock. The TF DNA-binding activities (DBA), which play key roles in regulating transcriptome, would impact the nuclear sub-proteome and in turn, the whole proteome; post-translational modifications, including phosphorylation and ubiquitylation, may also impart another layer of regulation. The complicated relationships among different layers raise many questions that remained to be answered, for instance: (1) how diurnal rhythmic phosphorylation of signaling transduction regulates the rhythm of TF DBA; (2) is there correlation between nuclear TF protein expression and TF DBA; (3) how diurnal rhythmic TF DBA correlates with the diurnal rhythm of downstream genes' transcription; (4) is there correlation between diurnal rhythms of mRNA expression and protein expression; and (5) how the ubiquitylation system controls the proteome oscillation. Answers to these questions will be informative in portraying a panoramic view of the circadian transcription regulation that governs the temporal switch of physiology in the mouse liver.

We previously developed an approach that enables the identification and quantification of endogenous TFs at the proteome scale. With a synthetic DNA containing a concatenated tandem-array of the consensus TFREs (catTFRE) as an affinity reagent, we can identify almost all expressed TFs in cell lines and tissues[12]. In this study, we employ this catTFRE approach to profile the dynamic of the TF sub-proteome during the circadian cycle of the mouse liver[13]. A total of 297 TFs are quantitatively identified, and 80 of which show circadian oscillations ($p < 0.1$). In addition to TF DBA, we also measure the nuclear sub-proteome, whole-liver transcriptome, whole-liver proteome, phosphorylation, and ubiquitylation patterns to build a TF DBA-centered multi-dimensional proteomics landscape of circadian regulation in the mouse liver. Four major biological processes, including immune response, fatty acid metabolism, cell cycles, and glucose metabolism, are found to be oscillating in this TF DBA centric view, facilitating our further understanding of the physiological regulations of the circadian clock in the liver.

## Results

**Multi-dimensional liver proteome of the circadian clock.** To map the molecular landscape of circadian clock using mouse liver as a model system, we took advantage of several current techniques to measure (1) TF DBA patterns (DBAP); (2) post-translational modifications (PTM) including phosphorylation and ubiquitylation; (3) transcriptome, and (4) protein abundance, including the nuclear sub-proteome and the whole proteome (Fig. 1a). We aimed to provide a multi-dimensional data set for the investigation and understanding of circadian clock system within the framework of a TF-centric paradigm.

**In-depth TF DBAP during circadian cycle of mouse liver.** We first profiled the diurnal rhythm of the TF DBA (Methods section; Supplementary Fig. 1a). Using the TF data set reported by Ravasi et al.[14], we identified 617 DNA-binding proteins (DBPs), among which, 297 were defined as TFs. We also detected 247 transcription co-regulators (TCs) (Fig. 1b; Supplementary Data 1). The same time points in the two consecutive circadian cycles showed high correlation coefficient with an average Pearson's $r$ of greater than 0.96 (Supplementary Fig. 1b). An average of 206 TFs were quantified for the 16 time points, ranging from 150 TFs in the ZT12 (Zeitgeber time) to 230 TFs in the ZT0. Oscillations of TF numbers as well as numbers of DBP and TC numbers were evident (Fig. 1c, d) during the consecutive circadian cycles.

**Diurnal oscillations of mouse liver transcription factors.** We grouped the ZT points into four time phases (TP) in 6 h intervals based on the oscillation patterns of the TF DBA. We then selected TFs whose DBA in any specific TP that were more than twofolds greater than that in the rest of the TPs. These TFs may perform specific functions in the specific TP and were therefore named as TP-specific TFs (Fig. 2a). GO/pathway analyses of TP-specific TFs identified the following enrichments: immune response in TP1, lipid metabolism in TP2, cell cycle in TP3, and glucose metabolism in TP4 (Fig. 2b), suggesting a clear division of labor for hepatic functions during the circadian cycle.

We then investigated diurnal rhythmicity of DBA of the TP-specific TFs. Diurnal rhythmicity was identified according to JTK_CYCLE[8] method (Methods), in which 159 DBPs, 80 TFs, and 59 TCs were found as diurnal rhythmic proteins ($p < 0.1$), accounting for 26%, 27%, and 24% of the total identifications in its functional group, respectively (Fig. 2c; Supplementary Data 1). Importantly, Bmal1, Clock, Cry2, and Bhlhe40 were found with significant diurnal rhythms. Interestingly, they were not found to be diurnal rhythmic in protein abundance from previous studies[15,16] (Fig. 2d). We also found that the circadian transcription activator protein complex, Clock/Bmal1, showed high synchrony in DBA (Pearson's $r$ 0.96), while the activity of the transcription repressor Cry was negatively correlated with the Clock/Bmal1 complex.

We identified the diurnal rhythmic TFs ($p < 0.1$) within TP-specific TFs, in which 11/48 TP-specific TFs in TP1, 8/33 TP-specific TFs in TP2, 2/12 TP-specific TFs in TP3, and 5/41 TP-specific TFs in TP4 were diurnal rhythmic, respectively. The GO/pathway enrichments of diurnal rhythmic TFs in each TP were highly consistent with the functions enriched by the TP-specific TFs, suggesting that the hepatic gene transcription and physiology functions are dominated by the diurnal rhythmic TFs (Fig. 2e; Supplementary Data 2).

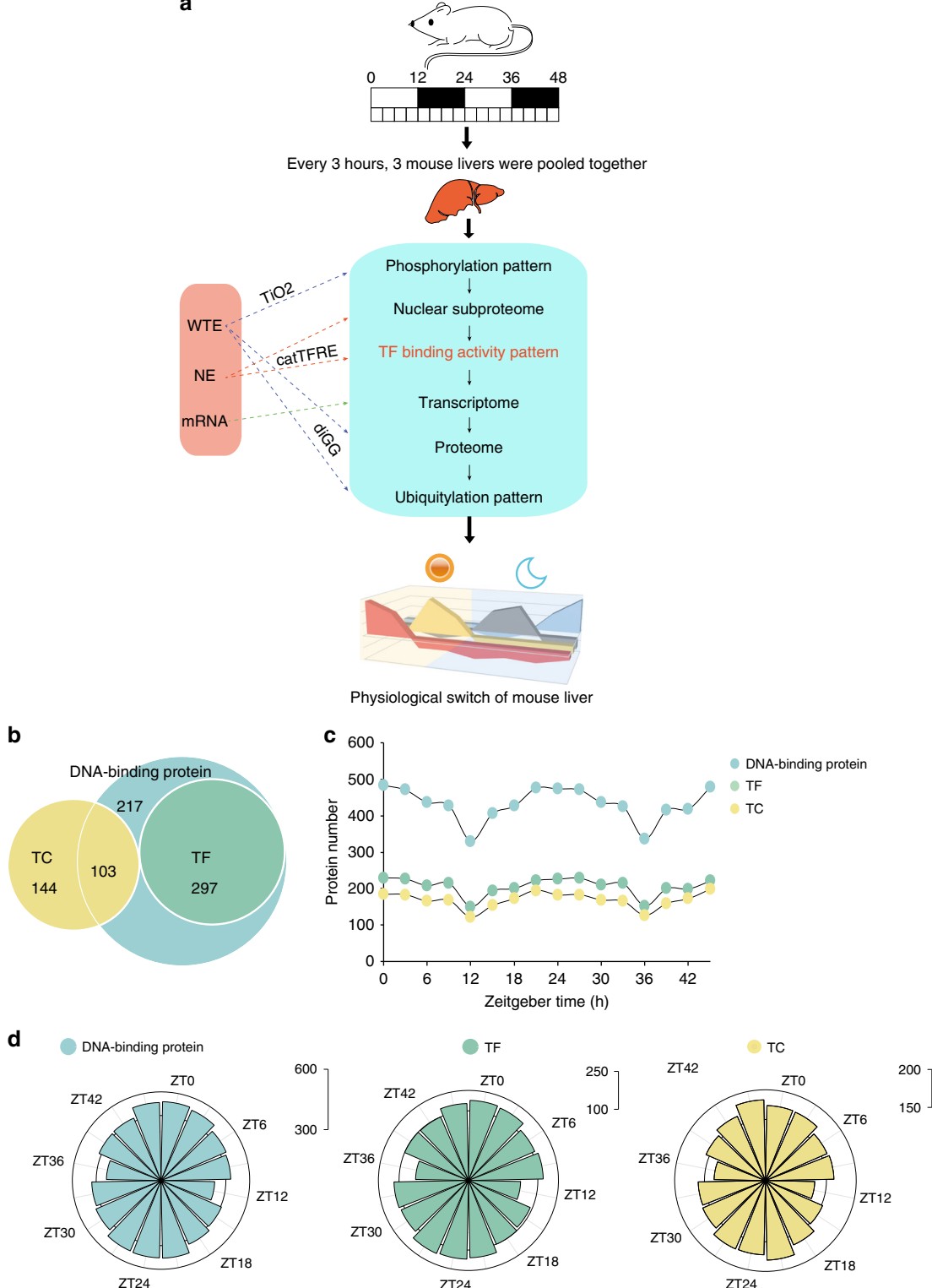

**Fig. 1** Transcription factor centered multi-omics of circadian clock in mouse liver. **a** Systematic workflow of the multi-omics experiments of the circadian clock in mouse liver. **b** Venn diagram illustrates the number of DBPs, TFs, and TCs detected in our deep TF DNA-binding activity profiling. **c**, **d** The number of DBPs, TFs, and TCs detected at each ZT point across two consecutive cycles

As previously reported, the NRs have been known as a typical diurnal rhythmic TF family on the mRNA level[17,18]. In our study, we detected 36 of the 49 NRs (Supplementary Fig. 2a, Supplementary Data 1), including core clock regulators such as Rev-erbα and Rorα, (Supplementary Fig. 2c) and 10 NRs that showed diurnal rhythm (Supplementary Fig. 2b). While the diurnal rhythms of the 7 NRs passed the JTK_CYCLE test, Mlr, Vdr, and Esr1 displayed single-pulse. Compared with the previous work on mRNA level conducted by the Nuclear Receptor Signaling Atlas (NURSA), we found that Pparσ, Trα, Rev-erbα, and Tr2 were diurnal rhythmic on both transcription level and DBA level, while Rorα, Lxrβ, Rxrβ were only diurnal

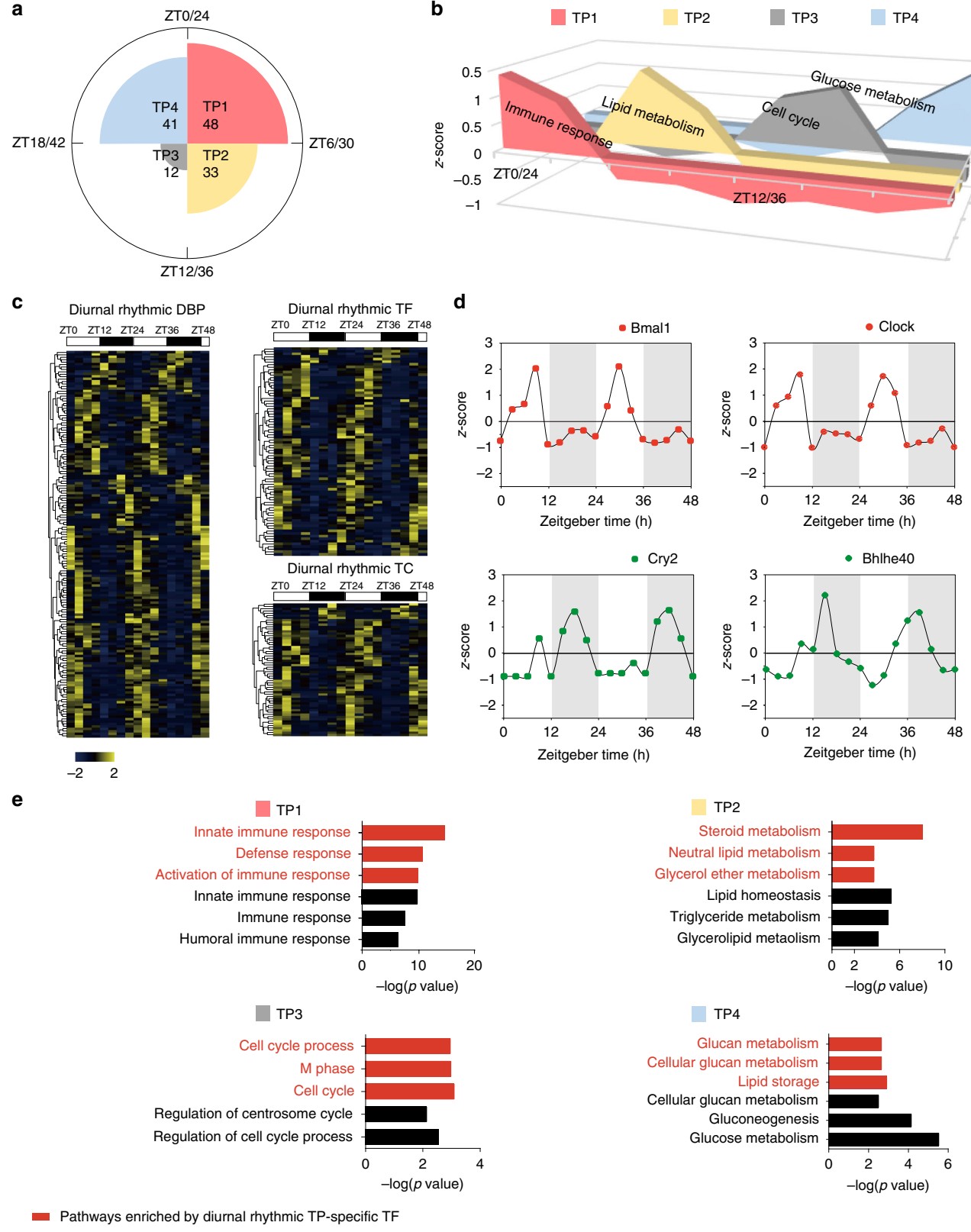

**Fig. 2** Diurnal oscillations of hepatic transcription factors. **a** Rose diagram shows the number of TP-specific TFs of each TP. **b** Area graph shows the normalized abundance of TP-specific TFs DNA-binding activity and the bioprocess enriched by TP-specific TFs. **c** Cluster heat maps of the diurnal rhythmic DNA-binding activity of DBPs, TFs, and TCs. The color bar indicates normalized $z$-scored iBAQ. The upper white to black bar indicates the 2 days' cycle. Daytime is shown in white, while nighttime is shown in black. **d** Temporal levels of the abundance of DNA-binding activity of Bmal1, Clock, Cry2, and Bhlhe40 in two consecutive cycles. $X$ axis represents the sampled time points, $Y$ axis represents the relative abundance of DNA-binding activity (normalized $z$-scored iBAQ) of each protein. **e** The GO bioprocesses enriched by diurnal rhythmic TP-specific TFs (red) and by all TP-specific TFs (black) of four TPs

rhythmic on the DBA level, probably because they functioned as a partner of heterodimer with regulatory genes involved in liver metabolism[19,20]. Vdr, which was not detected in the liver on mRNA level, was detected with a single-pulse in the ZT0 and ZT24 during the two consecutive cycles in this study (Supplementary Fig. 2a), highlighting the unique value of our TFRE approach.

**Diurnal rhythmic TF DBA was not originated from nuclear TF.** To determine whether the diurnal rhythms of TF DBA were originated from the diurnal rhythmic expression levels of the nuclear TFs, we measured the liver nuclear proteome. We identified 4038 (Supplementary Fig. 3a) proteins from purified nuclei in the two diurnal cycles including 105 TFs (Supplementary

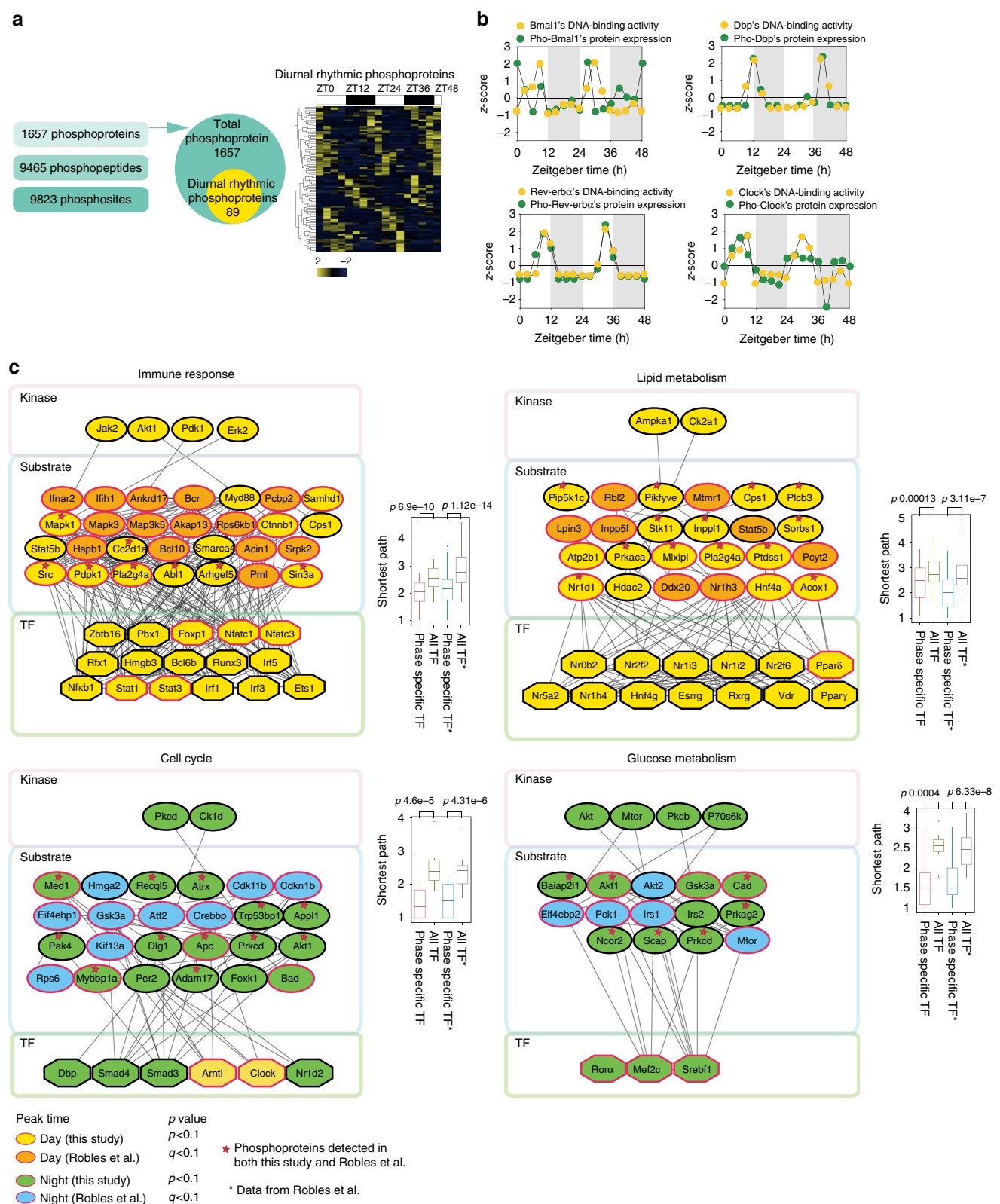

Data 3). While 79 were detected in the TFRE (DBA profiling) data set, only 10 TFs ($p < 0.1$) were diurnal rhythmic based on the JTK_CYCLE analysis (Supplementary Fig. 3b). The Pearson's correlation coefficient was low ($r = 0.179$) between TF DBA and TF protein abundance (Supplementary Fig. 3c). Consistent with the pervious study[10], Bmal1 and Clock did not demonstrate diurnal rhythmicity on the protein expression level. We then performed parallel reaction monitoring (PRM) and western blotting on several TFs protein expression level to validate these findings. Protein expression levels of TFs such as Bmal1/Arntl, pparδ, Rorα did not demonstrate oscillations throughout the circadian cycle (Supplementary Fig. 3d, e). Thus, pronounced diurnal oscillation of TF DBA is unlikely originated from oscillation of TF abundance during the circadian cycle.

**Diurnal rhythmic DBA was closely related with phosphorylation.** Phosphorylation plays an important role in TF DNA-binding[21,22]. We next carried out phosphorylation profiling following the TiO$_2$ enrichment[23] (Supplementary Fig. 4a). We identified 9465 phosphorylated peptides, representing 1657 phosphoproteins in the 16 time points across the diurnal cycles. Phosphorylation level of 89 proteins were judged as diurnal rhythmic ($p < 0.1$) (Fig. 3a; Supplementary Data 4). Of the 72 TFs in the phosphoproteome data set, 55 was identified in the DBA profiling, and 22 of them were diurnal rhythmic in DBA ($p < 0.1$), accounting for 40% (22/55) of the phosphorylated TFs. The high concordance between phosphorylation pattern and DBA of many TFs was highlighted by the master circadian clock TFs including Bmal1, Dbp, and Clock (Fig. 3b; Supplementary Fig. 4b, Supplementary Data 4).

We tried to classify the phosphoproteome into four TPs as we observed in the TF DBA, but failed to find clear functional enrichments. When we separated the phosphorylated proteins into two groups (daytime and nighttime group) based on their peak time, we found that the daytime group was enriched in immune response and lipid metabolism, whereas the nighttime group was enriched in the cell cycle and glucose metabolism (Supplementary Fig. 4c, d, Supplementary Data 5). These results suggest that the phosphorylation pattern provides lower time resolution than the TF DBA did in functional enrichments. We then constructed a computational model for the diurnal rhythmic signaling pathways using a recently published data[10] and used the kinase-substrates relationship database[24] to infer the kinases and used a protein–protein interaction database to find TFs that may be targeted by the kinases. As shown in Fig. 3c, the diurnal rhythmic expressed phosphoproteins turned out to be enriched in four bioprocesses: the immune response kinase-TF (K–T) axis mediated by Akt/Pdk1/Erk2–Stat1/Stat3/Nfkb1, and the lipid metabolism K–T axis mediated by Ampka1/Ck2a1–Nr2f2/Pparδ/Vdr at daytime, the cell cycle K–T axis mediated by Pkcd/Ck1d–Smad4/Smad3 and glucose metabolism K–T axis mediated by Mtor–Rorα/Mef2c/Srebf1 at nighttime.

**Nomination of the dominant rhythmic TFs.** As TF DBA may dictate downstream gene transcription, we set out to correlated the diurnal rhythmic mRNA oscillation with the perspective of TFs. We profiled the transcriptome during the circadian cycles and found that 2349 of the 11,120 identified mRNAs showed diurnal rhythm ($p < 0.05$), which include the core clock TFs Bmal1, Per1, and Dbp (Fig. 4a, b; Supplementary Data 6). Similar to the TF DBA, the rhythmic gene transcripts were enriched in the immune response, cell cycle, lipid metabolism, and glucose metabolism. However, the enrichment of these four bioprocesses were not apparent in a 4-TP rhythm, but in a 2-group rhythm (Fig. 4c; Supplementary Data 7), with lipid metabolic and immune processes in the daytime group, and cell cycle and glucose metabolic processes in the nighttime group. This pattern of behavior is similar to the phosphorylation but different from the TF DBA.

Next, we identified TFs that may control the diurnal rhythm of gene expressions in the mouse liver, and designated these TFs as "dominant rhythmic TFs" (DR-TFs). We reasoned that DR-TFs should exhibit diurnal rhythmic DBA, and also control the diurnal rhythmicity of their downstream target genes. To nominate DR-TFs, we used the TF and downstream TG (target gene) correlation database from the CellNet[25] to integrate the TF DBA and the transcriptome data. This analysis allowed us to identify 46 DR-TFs out of the 80 diurnal rhythmic TFs (Supplementary Data 8), including the well-characterized master clock proteins Bmal1, Clock, Dbp, Bhlhe40, and several other TFs that include Stat1, Mlxip, Mef2c, Cebpb (Fig. 4d). These DR-TFs may provide a framework to understand circadian regulation of transcription as exemplified in Fig. 4e. For example, the Stat1 and Stat3, which are the DR-TFs that peak in the daytime, control the transcription of immune-related genes such as Irf5 and Dhx58, which are also peaked in the daytime (Fig. 4e; Supplementary Fig. 8b).

We defined transcriptional "activators" as TFs whose DBA are positively correlated with transcripts of their target genes (Pearson's $r > 0.5$), and "repressors" whose activities are negatively correlated with transcripts of their target genes (Pearson's $r < -0.5$). It is interesting to note that DR-TFs peak in a TP tend to "activate" TGs enriched in the dominant bioprocess of this TP, and to "repress" TGs enriched in the dominant bioprocess of other TPs (Supplementary Fig. 5a, b, Supplementary Data 11).

**Comparison of diurnal rhythm of transcriptome with proteome.** Previous studies have demonstrated poor correlation between the transcriptome and the proteome[16,26,27]. We performed whole proteome profiling using the same batch of liver samples (three mice were killed every 3 h for 48 h)[28] (Supplementary Fig. 6a). We identified a total of 6780 proteins, in which 575 proteins were diurnal oscillating ($p < 0.1$), accounting for 8% of the total proteins (Fig. 5a; Supplementary Data 9). The GO bioprocess enrichment of diurnal rhythmic proteins showed the consistency of the dominant diurnal pathways, the gene transcription regulation and protein expression in, namely immune response, cell cycle, lipid

**Fig. 3** Diurnal phosphoproteome of mouse liver. **a** Numbers of phosphosites, phosphopeptides, and phosphoproteins detected in our diurnal phosphoproteome of mouse liver. The distribution of diurnal rhythmic phosphoproteins in the total phosphoproteins. Hierarchical clustering of the cycling phosphoproteins ordered by the phase of the oscillation. Values for each phosphoprotein at all analyzed samples (columns) are color code based on the intensities, low (blue) and high (yellow) z-scored normalized iBAQ. The upper white to black bar indicates the 2 days' cycle. Daytime is shown in white, while nighttime is shown in black. **b** Temporal abundance of DNA-binding activity of Bmal1, Clock, Dbp, and Nr1d1/Rev-erbα and their correlated abundance of their phosphorylated forms. X axis represents the sampled time points, Y axis represents the z-scored abundance. **c** Time-resolved map of signaling cascades regulated by diurnal changed phosphoproteins. Proteins (kinases, phosphoproteins, TFs) were colored based on their peak time, daytime-peaked proteins are yellow (detected in this study) or orange (detected in the study by Robles et al.[10]), while nighttime-peaked proteins were blue (detected in the study by Robles et al.[10]) or green (detected in this study). Diurnal rhythmic proteins (this study: JTK_CYCLE $p < 0.1$, Robles et al.[10]: $q$ (adjusted $p$) < 0.1) were shown with red border. Box plots show the path from substrates to pathway-specific TFs were shorter than the path from substrates to all detected TFs (pair tailed Student's $t$ test $p < 0.05$). For the box plot, the bottom and top of the box are the first and third quartiles, and the band inside the box is the median of the paths between TFs and substrates

metabolism, and glucose metabolism (Fig. 5a, b; Supplementary Fig. 6c), (Supplementary Data 10). Specifically, the core proteins of innate immune response, the key enzymes of the glucose metabolism were dominant during the daytime, while the key enzymes in the lipid metabolism and TCA cycle were upregulated during the nighttime (Supplementary Fig. 6d).

In the proteome data set, we found that, while the mean coefficient variation during the circadian cycle is 0.68, 30% of proteins have coefficient variations larger than 0.9 (Supplementary Fig. 6b). We then compared the CVs (coefficient variation) between proteome and transcriptome, and found that genes whose transcriptome varies greatly than proteome were mainly

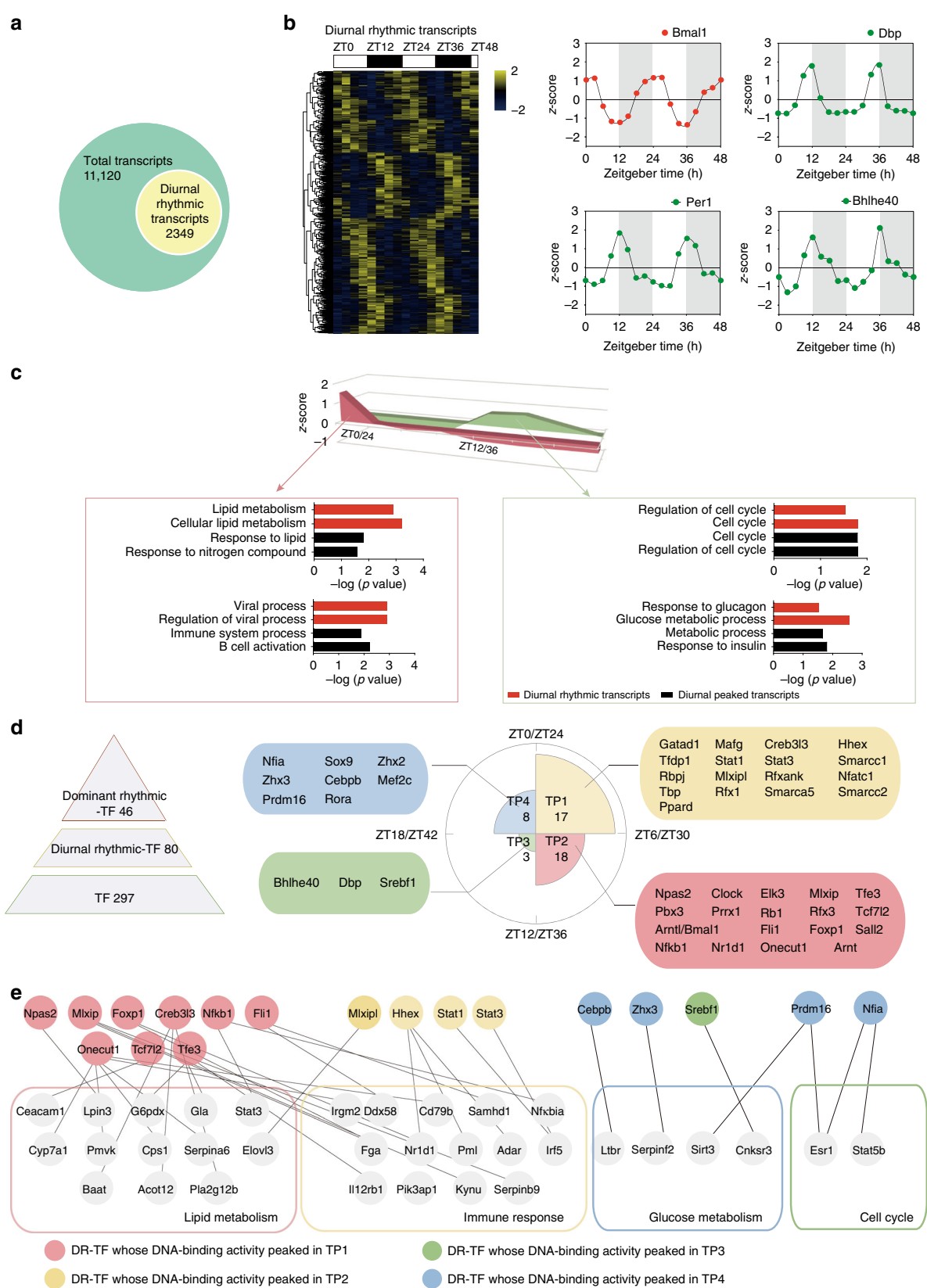

enriched in metabolism(Supplementary Fig. 7a). The correlation coefficients between transcriptome and proteome in all 16 time points were between 0.5 and 0.6 (Supplementary Fig. 7b), which is consistent with previous reports[29,30]. Of the 2349 diurnal rhythmic gene transcripts, 1256 of them were identified in the proteome, and only 133 of them displayed diurnal rhythm on both mRNA ($p < 0.05$) and protein levels ($p < 0.1$). The diurnal rhythmicity of 2216 mRNAs and 442 proteins were detected in mRNA or protein only. Interestingly, transcript-specific diurnal rhythmic genes were enriched in RNA processing/splicing, while protein-specific diurnal rhythmic genes were enriched in toxin transport and lysosomal transport (Fig. 5c). These results suggest that the diurnal rhythm of transcriptome is not necessarily translated to the proteome, and the diurnal rhythm can be better understood by measuring both transcripts and proteins.

We also nominated "dominant diurnal rhythmic TFs (DR-TFs)" from the protein profiling data using the similar criteria as we did with the transcriptome data set. Surprisingly, 41 "DR-TFs" were found using the proteome data set, covering 40 of the 46 DR-TFs identified from the transcriptome data set (Supplementary Fig. 8a, Supplementary Data 8). A substantial overlap of TGs of the DR-TFs at proteome level were also observed (Supplementary Fig. 8c). Together, these results indicated that, while the overall diurnal rhythm cannot be directly translated from mRNA to protein, DR-TFs derived from the transcriptome and the proteome data sets are more consistent, suggesting that TF DBA may be the major driver in governing circadian regulation in mouse liver.

**The dynamic ubiquitylated pattern during the circadian cycle.** To gain further insight into the circadian clock in mouse liver, we measured another PTM—ubiquitylation to survey another dimension of the protein landscape during the circadian cycle (Fig. 5d). We identified 3424 ubiquitylated sites on 1144 proteins during one circadian cycle (Fig. 5e; Supplementary Data 12). The number of the detected ubiquitylated proteins appeared to be more in daytime than in nighttime, while the number of total proteins detected in the proteome was similar (Fig. 5e). Clustering analyses also revealed that ubiquitylated proteins that peaked in the daytime and were enriched in immune response and lipid metabolism, while the ubiquitylated proteins that peaked in the nighttime were enriched in glucose metabolism (Fig. 5f).

In the ubiquitylation data set, it appeared that 5% of the diurnal rhythmic TFs, particularly DR-TFs, such as Stat1 and Stat3, were ubiquitylated, while 2% of the non-diurnal-rhythmic TFs were ubiquitylated. We also compared the ubiquitylation patterns of their TGs. We found that 8% of the TGs of the diurnal rhythmic TFs were ubiquitylated, while 6% of the TGs of the non-diurnal rhythmic were ubiquitylated (Fig. 5g). Additionally, the number of ubiquitylated TGs of the diurnal rhythmic TFs was significantly higher than the number of ubiquitylated TGs of the randomly selected TFs (Fig. 5h), indicating that target gene products (GPs) were also further regulated by ubiquitylation, in addition to their regulation by TFs. The GO analysis revealed that the ubiquitylated diurnal rhythmic proteins were significantly enriched in the major diurnal rhythmic pathways, including

glycolysis/gluconeogenesis, fatty acid degradation, PPAR signaling, but the enrichment was not observed in non-ubiquitylated diurnal rhythmic proteins (Fig. 5i). Proteins involved in the innate immune response, which include the receptors (Il23r, Fcer1g, Fcgr2b et al.), the enzymes and adaptors (Rac1, Trim25, Ikk, Trafd1 et al.), and the effector TFs (Stat1 and Stat3) were all ubiquitylated. The same phenomenon was also found in the fatty acid metabolic process (Fig. 5j).

We also found consistency in the diurnal rhythmicity of protein expression and their ubiquitylation level, including Trim25, Iκb, Irgm1 in the immune pathway, Me1, Slc27a5, and Apoa2 in the fatty acid metabolic pathway (Supplementary Fig. 9). These results collectively suggest that ubiquitylation is another important mechanism in the regulation of diurnal rhythm in addition to transcription regulation.

**The diurnal rhythmic regulatory network of the Med complex.** In addition to TF, co-regulator is another class of proteins that play important roles in transcription regulation. The multi-subunit Mediator complex appeared to be a hub of transcription regulation[31]. In our TFRE data set, we detected 19 of the 30 components of the Med complex and found that many of them exhibited diurnal oscillating pattern. For instance, Med1 and Med27 peaked in the daytime, while Med12 and Med24 peaked in the nighttime (Fig. 6a; Supplementary Data 1). DBA of Med1 and Med27 showed strong diurnal rhythmic oscillation ($p < 0.1$). This observation was confirmed by manually extracting XIC (extracted ion chromatogram) and western blotting (Supplementary Fig. 10a and b).

An interaction between Med1 and Clock has been reported previously[32]. We then calculated the correlation coefficient between components of Med complex with the diurnal rhythmic TFs. The diurnal rhythmic DBA of many TFs, including Bmal1, Clock, Naps2, Elf2, and Nf-κb1 were significantly correlated with that of Med1 (Supplementary Fig. 10c, Supplementary Data 1). It is interesting to note that TFs whose DBA peaked in different time of the clock tends to correlate with different components of the Med complex (Fig. 6b), suggesting that different components of the Med complex may be recruited by the diurnal rhythmic TFs during the circadian clock.

We constructed a computational model for a hierarchal network centered around the Mediator complex (Fig. 6c). We focused on the four essential diurnal rhythmic biological processes, namely the immune response, lipid metabolism, cell cycle and glucose metabolism, to implicate the role of the Mediator complex in these biological processes. We found that the correlation between the Med complex and pathway-specific TFs (TFs enriched in a pathway or bioprocess) is significantly higher than that with all detected TFs (average Pearson's $r$ 0.78 vs 0.19, $p$ value 1.6e−6) (Fig. 6c). Furthermore, the correlations were even higher in the pathway's dominant TP as defined by TFRE analysis than in the other phases, suggesting that the Med complex serves as a gene regulatory hub in the diurnal regulation network.

**The diurnal switch of the proteome of Kupffer cells.** Liver is an organ with predominant innate immunity, which can specifically

**Fig. 4** Diurnal transcriptome of the mouse liver. **a** Distribution of diurnal rhythmic transcripts. **b** Hierarchical clustering of the cycling transcripts ordered by the time of the oscillation. Values for each transcript at all analyzed samples (columns) are color code based on the intensities, low (blue) and high (yellow) z-scored normalized FPKM. The upper white to black bar indicates the 2 days' cycle. Daytime is shown in white, while nighttime is shown in black. Temporal abundance of transcripts of Bmal1, Dbp, Per1, and Bhlhe40 in 2 consecutive cycles. X axis represents the sampled time points; Y axis represents z-scored FPKM. **c** Area graph shows the normalized abundance of transcripts that peaked in different time of the day (red: daytime, green: nighttime). Bar plots shows the GO terms enriched by diurnal peaked transcripts (black), and GO terms enriched by diurnal rhythmic transcripts (JTK_CYCLE, $p < 0.1$) (red). **d** The hierarchical pyramid shows the number of TFs, diurnal rhythmic TFs, transcriptome nominated DR-TFs. The Rose diagram shows the number of transcriptome nominated DR-TFs of each time phase, TFs were colored based on their peaked phase, (TP1 shows in yellow, TP2 shows in red, TP3 shows in green, TP4 shows in blue). **e** The regulation network of DR-TFs to their TGs (target genes), TFs were colored based on their peak phase

detect infection through pattern-recognition receptors (PRRs) that recognize specific structures, called pathogen-associated molecular patterns (PAMPs)[33]. Toll-like receptor (TLR) signaling is the dominant PAMPs of the liver[34]. We detected large number of proteins in TLR signaling pathways, including the receptors (Tlr3, Tlr6), the adaptor (Myd88), signal transducers

(Irf3, 5, 6, and 9), TFs (Nf-κb2) and the effector cytokines (Il23r, Tnfα). Their abundance varied diurnally with peak phase in the daytime and valley phase in the nighttime. Two Tnfα inducible proteins, Tnfaip3 and Tnfaip8, were expressed higher in the nighttime, suggesting a temporal regulation of the TLR signaling (Supplementary Fig. 6d). The expression level of the Nf-κbib was

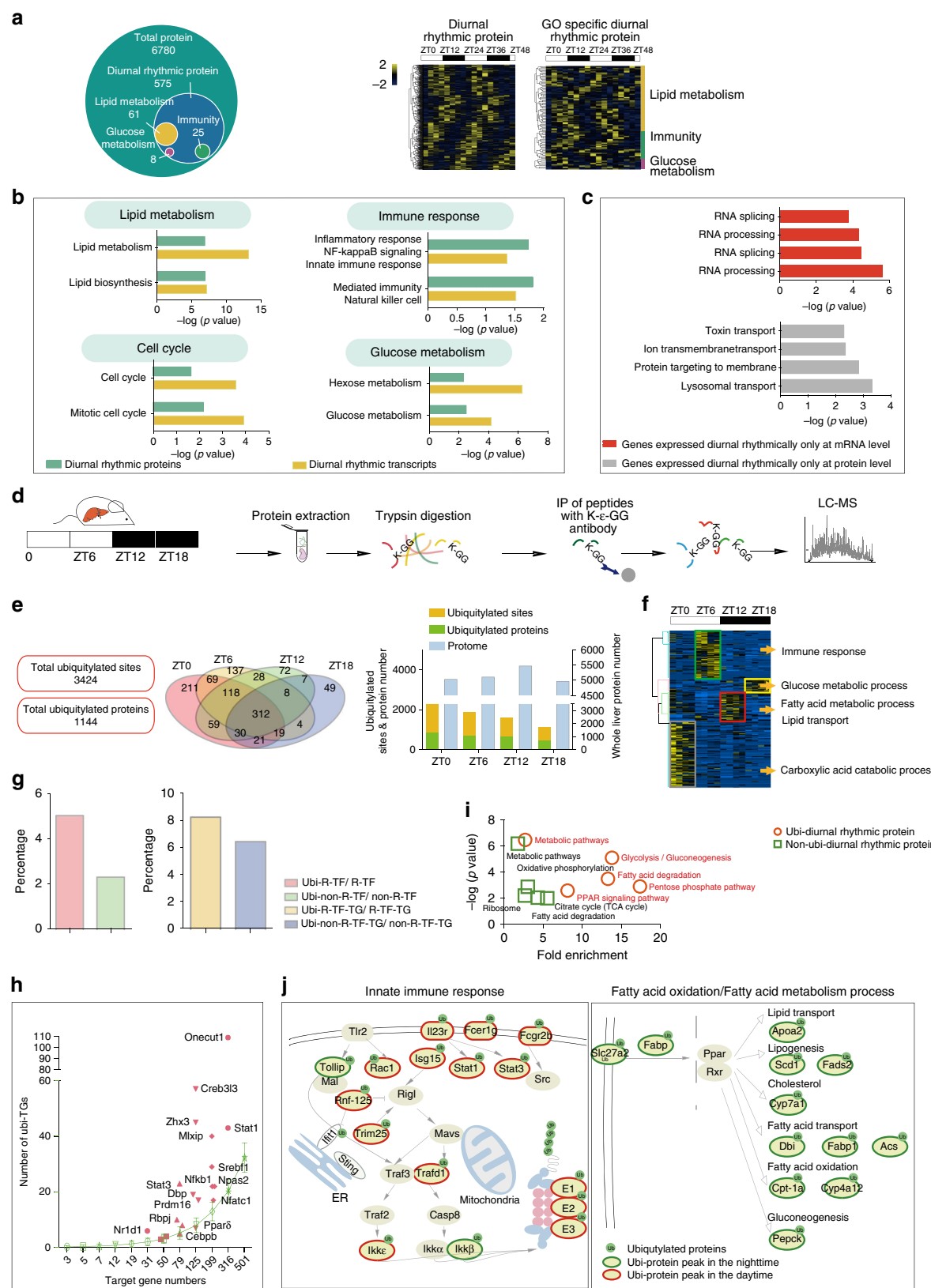

negatively correlated with that of Nf-κb2 in the circadian clock, consistent with its role as an Nf-κb inhibitor.

Since Kupffer cells (KCs), known as liver-resident macrophages, may be the cell-type responsible for the innate immunity in the liver[35], we asked whether the daytime and nighttime variations of immune response in the liver was attributable to variations in KCs. To this end, we isolated primary KCs from liver at four different time points of the circadian clock (ZT0, ZT6, ZT12, and ZT18), and performed MS-based proteome analysis (Fig. 7a). Interestingly, the number of KCs seemed to vary during the circadian cycle, peaking at ZT6 (1.8e6/liver), and falling at ZT12 (4.56e5/liver) (Fig. 7b). Among the 4684 identified KC proteins, 690 showed diurnal rhythm ($p < 0.1$) (Fig. 7c; Supplementary Data 13); however, the diurnal rhythmicity of only 61 of them were observed in the whole-liver proteome, suggesting that the cell-type-resolved KC proteome provided more information than the whole-liver proteome (Fig. 7d). Further analyses indicated that as many as 1078 proteins showed strong oscillation between daytime and nighttime with fold changes >5, although they failed to pass the JTK_CYCLE test. They included many well-characterized immune-related proteins, such as Tmem173 and Mavs. A selected group of proteins with diurnal expression differences between the KC proteome and the whole-liver proteome were confirmed by MS measurement and western blotting (Supplementary Fig. 11). The GO analysis demonstrated that the diurnal differential expressed proteins in the KC were much more significantly enriched in the immune response pathways than the whole-liver proteome (Fig. 7e). Furthermore, the abundance of the major immune response pathway appeared to peak in the daytime, as exemplified by the major components of the TLR pathway including Tlr4, Myd88, Irak4, and Tak1 (Fig. 7f, g).

We further constructed interaction networks between KC and whole-liver proteins around the circadian clock. The connections between KC and whole-liver proteins in immune response pathway were more dominant in the daytime than the nighttime ($p = 1.5e68$), while the opposite phenomenon was observed for the metabolism process, and more connections between KC proteins and whole-liver proteins were observed in the nighttime ($p = 1.5e18$) than in the daytime (Fig. 7h, i). These results suggested a potential cooperation between KC and the liver organ.

We then employed a TLR-induced liver injury model to test whether the diurnal rhythm of innate immune has any biological consequences using LPS treatment[36,37]. We administrated LPS/D-GalN to the mice in two ZT points, ZT0 and ZT12, and then measured the survival rate, serum ALT (alanine aminotransferase) and AST (aspartate aminotransferase) levels and histology of the liver. As shown in Fig. 8a, the group treated with LPS at ZT0 had lower survival rate, higher ALT/AST levels, and more sever liver damage compared to the group that were treated at ZT12. These results suggest that the immune systems in the liver may have differences during the daytime and nighttime and is consistent with reports that temporal differences of macrophages were found in serum, spleen, and lymph nodes[38–40].

We next injected the LPS/D-GalN to the mice in two ZT points, ZT0 and ZT12, then isolated KCs 2 h after co-injection, and profiled their proteomes (Fig. 8b). We observed that the abundance of proteins of the innate immune pathway were higher in the daytime than in the nighttime (Fig. 8c). We then focus on the Tlr4 signal transduction pathway, which directly responds to LPS. We found that abundance of proteins including Myd88, Nf-κb, Irak4 were highly expressed in the daytime than at nighttime (Fig. 8c), indicating that the Tlr4 signaling transduction pathway were elevated and may be more sensitive to LPS during the daytime.

## Discussion

As the major metabolic organ, liver plays an important role in regulating metabolism homeostasis around the circadian clock. A number of transcriptome and metabolome studies on circadian clocks have illustrated the remarkable roles of circadian clock on cellular and organismal physiology[41,42], but relatively little is known about the temporal regulation of gene expressions and its driving force at the proteome level. Two previous studies by Daniel Mauvoisin et al.[15] and Robles et al.[16] quantified 5000 and 3000 proteins, respectively. These studies revealed the dynamics of diurnal liver proteome. However, due to technical limitations, these studies failed to identify the core clock transcription factors[15,16].

In this study, we employed catTFRE[12] to profile the dynamics of TF sub-proteome during circadian cycle. As the result, 80 out of 297 identified TFs were determined as diurnal rhythmic TFs (JTK_CYCLE, $p < 0.1$) during two circadian cycles. Bioinformatic analysis indicated that the diurnal rhythmic TFs could be grouped into four major pathways, namely the immune response, glucose metabolism, fatty acid metabolism and the cell cycle, based on oscillation patterns of their DBA.

The quantification method for the proteome now is mostly label-free quantification (LFQ). Compared with label-based MS quantification, the LFQ can quantify unlimited number of samples for comparison and may also offer higher dynamic range to detect low abundance proteins[43], the LFQ using the iBAQ

**Fig. 5** Diurnal proteome of mouse liver. **a** Distribution of pathway-specific diurnal rhythmic proteins. Yellow, green, and purple cycle represent the number of proteins enriched in lipid metabolism, immunity, and glucose metabolism. Hierarchical clustering of the total (left) and GO specific diurnal rhythmic proteins (right) ordered by the phase of the oscillation. **b** The bar plot shows the major GO terms enriched by diurnal rhythmic transcripts (yellow), and diurnal rhythmic proteins (green). **c** The bar plot shows the GO terms enriched by the genes whose diurnal rhythm were inconsistent in transcript and protein level. **d** Systematic workflow of the ubiquitylation proteome during the circadian cycle in mouse liver. **e** The total number of ubiquitylated sites and proteins, and the number of ubiquitylation proteins detected at different ZT points. Bar plot shows the comparison between the numbers ubiquitylated proteins with the number of whole-liver proteins detected at different ZT points. **f** Hierarchical clustering of the ubiquitylated proteins ordered by the phase of the oscillation. **g** Bar plots show the percentage ratio between ubiquitylated rhythmic TFs and rhythmic TFs versus the percentage ratio between ubiquitylated non-rhythmic TFs and non-rhythmic TFs (left). The percentage ratio between ubiquitylated rhythmic TFs' TGs and rhythmic TFs' TGs versus the percentage ratio between ubiquitylated non-rhythmic TFs' TGs and non-rhythmic TFs' TGs (right). **h** The number of ubiquitylated TGs of the diurnal rhythmic TFs versus the number of ubiquitylated TGs of the randomly selected TFs. **i** Scatterplot shows statistically enriched GO/KEGG pathways by ubiquitylated diurnal rhythmic proteins versus pathways enriched by non-ubiquitylated diurnal rhythmic proteins. **j** Systematic overview of signal transduction participated by ubiquitylated proteins. Ubiquitylated proteins were colored based on their peak time. Ubiquitylated proteins peaked in the daytime were shown with red border, Ubiquitylated proteins peaked in the nighttime were shown with green border. For all the Hierarchical clustering heatmap, values for each protein at all analyzed samples (columns) are color code based on the intensities, low (blue) and high (yellow) z-scored normalized iBAQ. The upper white to black bar indicates the diurnal cycle. Daytime is shown in white, while nighttime is shown in black

algorism may suffer from lower accuracy and in general was able to quantify a change of more than a factor of 2, which was adequate for the characterization of many biological processes including circadian rhythm studies published previously[10,11].

We aimed to provide multi-dimensional proteomics data sets to portray the landscape of circadian clock in the mouse liver. While centered at TF DBA, our data sets also contain phospho-proteome, ubiquitylation proteome, nuclear proteome, whole

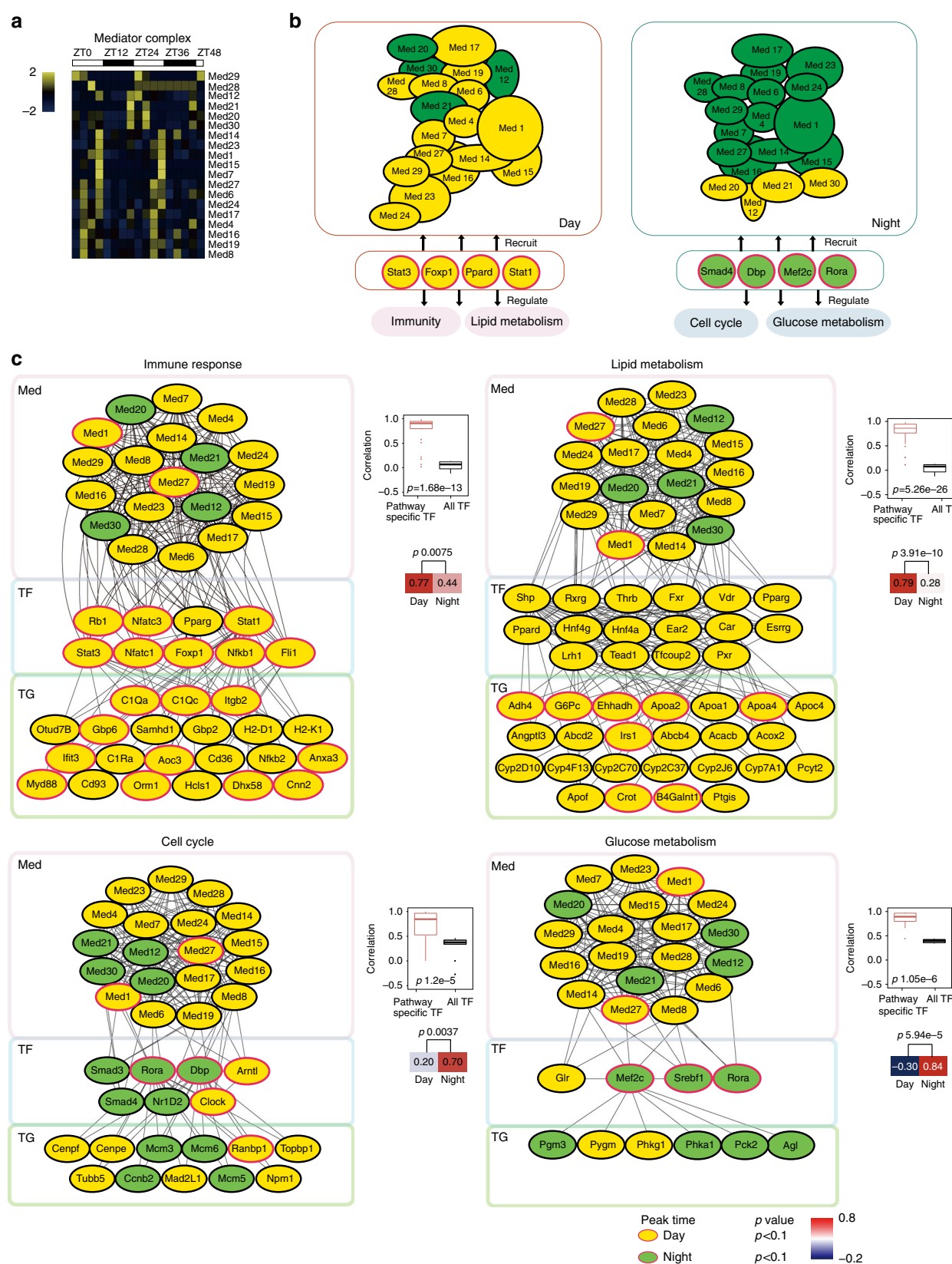

proteome, and the whole transcriptome measured from the same batch of samples. These data sets would allow other researchers in the field to perform in-depth mining of different layers of regulation of the circadian clock, which should lead to eventual construction of the hierarchal circadian rhythmic regulation network.

In summary, we have built a TF DBA-centered multi-dimensional proteomics landscape to illustrate the hierarchal gene expression network in the circadian clock of mouse liver. Our data sets include TF DBA, protein phosphorylation, ubiquitylation, transcriptome, nuclear protein profiling, and whole proteome profiling as well as the whole proteome of the Kupffer cells. We offered the most comprehensive data sets of the diurnal rhythm in the mouse liver by far and provided the richest data resource for the understanding of mouse liver physiology around the circadian clock.

## Methods

**Animals and tissue collection.** Eight-week-old C57BL/6, SPF (specific pathogen-free) male mice (weight 20–25 g) were housed in pathogen-free, temperature-controlled environment, scheduled with 12–12 h light–dark cycles. All mice were maintained with free access to food and water for 2 weeks. To collect tissues, three mice were killed every 3 h for 48 h. We grouped animals randomly, to ensure that the investigator was blinded to the group allocation during the experiment. This study is under the guidelines of the animal care regulations of Fudan University, and received ethical and scientific approval from Fudan University.

**Kupffer cell isolation and protein extraction.** Normal eight-week-old C57BL/6 male mice were housed, and scheduled with 12–12 h light–dark cycles for 2 weeks. At each time point three mice were used for KCs' isolation. We used two-step liver perfusion digestion in situ followed the previously described protocol[44]. KCs were first isolated simultaneously with LSECs using collagenase-based density gradient centrifugation. Cells between 11.2% and 17% in optiprep$^{TM}$ density gradient were carefully collected. Then cells were labeled with phenotypic markers (FITC-conjected CD11b and PE-conjected F4/80) and purified with FACS analysis (BD Biosciences, Franklin Lakes, NJ). Collected KCs were then quickly frozen with liquid Nitrogen and transferred to −80 °C refrigerator for storage.

**Protein sample preparation.** Three mice livers collected at each zeitgeber time point were pooled together to extract nuclear protein and whole-tissue protein.

Nuclear extractions: 0.3 g pooled tissues were homogenized with 800 μl Cytoplasmic Extraction Reagent I (CER I) buffer (NE-PER™ kit Thermo Fisher Scientific), and 8 μl protease inhibitor (Pierce™, Thermo Fisher Scientific). Nuclear proteins were extracted following the protocol provided by the manufacturer. Protein concentrations were measured using Bradford method (Eppendorf Biospectrometer).

Whole-tissue protein extractions: An aliquot of 0.1 g pooled tissues were lysed with 400 μl urea lysis buffer (8 M urea, 100 mM Tris-HCl pH 8.0), 4 μl protease inhibitor (Pierce™, Thermo Fisher Scientific) was added to protect protein from degradation and protein concentrations were measured using Bradford method (Eppendorf Biospectrometer).

KCs protein extractions: In total, 100 μl cell pellets were lysed with 400 μl urea lysis buffer (8 M urea, 100 mM Tris-HCl pH 8.0), 4 μl protease inhibitor (Pierce™, Thermo Fisher Scientific) was added to protect protein from degradation and protein concentrations were measured using Bradford method (Eppendorf Biospectrometer).

**CatTFRE pull-down and trypsin digestion.** We referred to TF-binding database JASPAR to select consensus TFREs for different TF families. To design the catTFRE construct, we used 100 selected TFREs and placed two tandem copies of each sequence with a spacer of three nucleotides in between, resulting in a total

DNA length of 2.8 kb. An aliquot of 1 mg NEs (nuclear extractions) were then performed catTFRE pull-down. 2.8 kb catTFRE DNA and biotinylated catTFRE primers were synthesized by Genscript (Nanjing, Jiangsu Province, China). In total, 3 pmol biotinylated DNA pre-bound to Dynabeads (M280 streptavidin, Thermo Fisher scientific), and then incubated with NEs in 4 °C for 2 h. After incubation, the supernatant was discarded, and the NETN buffer (100 mM NaCl, 20 mM Tris-HCl, 0.5 mM EDTA, and 0.5% [vol/vol] Nonidet P-40) and PBS was used to wash the Dynabeads beads twice respectively. An aliquot of 20 μl of 2× SDS (sodium dodecyl sulfate) loading buffer was used to re-suspend beads. The resuspended beads were then boiled at 95 °C for 5 min. As for SDS-PAGE separation, samples were stained with Coomassie Brilliant Blue, and sliced into six bands equally according to the molecular weight ranges, followed by in-gel digestion overnight at 37 °C with trypsin, referring to the protocol described before[12].

**Protein trypsin digestion and first dimension RPLC.** For whole-tissue proteome, and nuclear sub-proteome, KCs sub-proteome 100 μg whole-tissue proteins were digested by the FASP procedure[45]. Namely, the protein samples were supplemented with 1 M dithiothreitol (DTT) to a final concentration of 5 mM and incubated for 30 min at 56 °C, then added iodoacetamide (IAA) to a 20 mM final concentration, and incubated in the dark at room temperature. After half an hour incubation, samples were added 5 mM final concentration of DTT and keep in dark for another 15 min. After these procedures, protein samples were loaded into 10 kD Microcon filtration devices (Millipore) and centrifuged at 12,000 × g for 20 min and washed twice with Urea lysis buffer (8 M Urea, 100 mM Tris-HCl pH 8.0), twice with 50 mM NH$_4$HCO$_3$. Then the samples were digested using trypsin at an enzyme to protein mass ratio of 1:25 overnight at 37 °C. Peptides were extracted and dried (SpeedVac, Eppendorf).

As for the whole-liver proteome, the digested peptide then performed first dimension RPLC before LC-MS/MS. The dried peptides were loaded into a homemade Durashell RP column (2 mg packing (3 μm, 150 Å, Agela) in a 200 μl tip), then eluted sequentially with nine gradient elution buffer which contains mobile phases A (2% acetonitrile (ACN), adjusted pH to 10.0 using NH$_3$.H$_2$O) and 6%, 9%, 12%, 15%, 18%, 21%, 24%, 30%, 35% mobile phase B (98% ACN, adjusted pH to 10.0 using NH$_3$.H$_2$O). The nine fractions then were combined into six groups (6% + 24%, 9% + 30%, 12% + 35%, 15%, 18%, 21%) and dried under vacuum for sub-sequential MS analysis.

**Phosphoproteome sample preparation.** A total of 1 mg whole-liver protein lysates were digested with trypsin for the TiO$_2$ enrichment. To the digested peptides, 0.25 ml binding buffer (80% ACN, 5% trifluoroacetic acid (TFA (Sigma-Aldrich)), and 1 M lactic acid (Sigma-Aldrich)) was added, peptides were mixed at room temperature for 1 min at 2000 rpm, cleared by centrifugation, and transferred to a clean 0.5 ml tube. TiO$_2$ beads were subsequently added to peptides at a ratio of 4:1 beads/protein, and incubated at room temperature for 30 min on rotor with middle speed. Beads were subsequently pelleted by centrifugation for 2 min at 2000 × g, and the supernatant (containing non-phosphopeptides) was collected to repeat the enrichment for twice and then discarded. Beads were suspended in wash buffer (80% ACN and 5% TFA), transferred to a clean 0.5 ml tube, and washed a further four times with 0.5 ml wash buffer. After the final wash, beads were suspended in 0.05 ml wash buffer, three batch beads were all transferred onto a 0.2 ml StageTip with two pieces of C8, and centrifuged for 3 min at 500 × g or until no liquid remained on the StageTip. Bound phosphopeptides were eluted two times with different gradient of elution buffer (0%, 3%, 6%, 9%, 12%, 40% ACN, and 15% NH$_4$OH) and collected by centrifugation into six 0.5 ml tube, then combined them into three tubes (0% + 40%, 3% + 12%, 6% + 9%). Peptide samples were concentrated in a SpeedVac. Peptides were resuspended in buffer containing 5% MeOH and 10% FA for liquid chromatography-tandem mass spectrometry (LC-MS/MS) analysis.

**LC-MS/MS analysis.** Peptides from catTFRE tandem in-gel digestion were detected by Q Exactive Plus (Thermo Fisher Scientific) and peptides used for detecting proteome, nuclear proteome, phosphoproteome, and ubiquitylation proteome and KC sub-proteome were detected by Orbitrap Fusion Lumos (Thermo Fisher Scientific).

---

**Fig. 6** The diurnal rhythm of the Mediator complex. **a** Hierarchical clustering of the proteins ordered by the phase of the oscillation. Values for each protein at all analyzed samples (columns) are color code based on the intensities, low (blue) and high (yellow) z-scored normalized iBAQ. The upper white to black bar indicates the 2 days' cycle. Daytime is shown in white, while nighttime is shown in black. **b** Modular structure of Mediator complex recruited by diurnal rhythmic TFs that regulated different bioprocess, subunits colored in yellow or green represent subunits' high or low DNA-binding activity, respectively. **c** The regulation network of Mediators, TFs, and TGs in four major pathways. For each pathway, the network on the left shows the TFs recruited components of Mediator to regulate TGs' expression, proteins (TFs, components of Mediator, TGs) were colored based on their abundance (proteins peaked in the daytime are yellow, in the nighttime are green), and diurnal rhythmic ones were shown with red border (JTK_CYCLE p < 0.1). The upright box plot shows the correlation between pathway-specific TFs with Mediators is higher than the average correlation between Mediators with total TFs (pair tailed Student's t test p < 0.05). For the box plot, the bottom and top of the box are the first and third quartiles, and the band inside the box is the median of the correlation between TFs and Mediators. The Heat map on the down-right shows the diurnal difference of the correlation between pathway-specific TFs with Mediators (pair tailed Student's t test p < 0.05)

Q Exactive Plus LC-MS/MS analyses were performed on an Easy-nLC 1000 liquid chromatography system (Thermo Fisher Scientific) coupled to an Q-Exactive Plus via a nano-electrospray ion source (Thermo Fisher Scientific). The peptides from in-gel trypsin digestion were dissolved with 10 μl loading buffer (5% methanol and 0.2% formic acid), and 5 μl was loaded onto a 360 μm I.D. × 2 cm, C18 trap column at a maximum pressure 280 bar with 12 μl solvent A (0.1% formic acid in water). Peptides were separated on 150 μm I.D. × 14 cm column (C18, 1.9

μm, 120 Å, Dr. Maisch GmbH) with a linear 5–35% Mobile Phase B (ACN and 0.1% formic acid) at 600 nl/min for 75 min. The MS analysis was performed in a data-dependent manner with full scans (m/z 300–1400) acquired using an Orbitrap mass analyzer at a mass resolution of 70,000 at m/z 400. The top twenty precursor ions were selected for fragmentation in the HCD cell at normalized collision energy of 30%, and then fragment ions were transferred into the Orbitrap analyzer operating at a resolution of 17,500 at m/z 400. The automatic gain control (AGC)

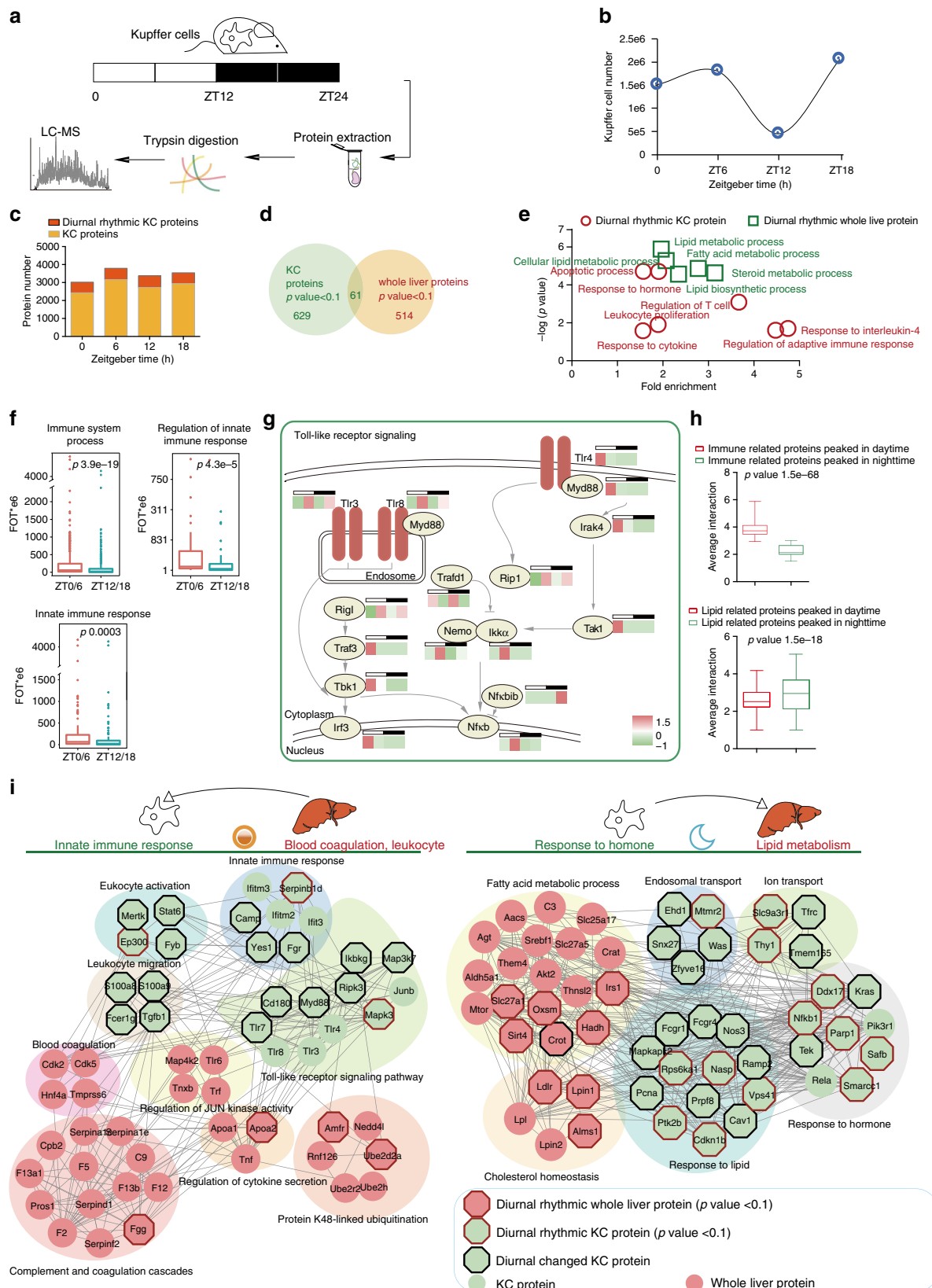

for full MS was set to 3e6, and that for MS/MS was set to 5e4, with maximum ion injection times of 20 and 60 ms, respectively. The dynamic exclusion of previously acquired precursor ions was enabled at 18 s.

Orbitrap Fusion Lumos LC-MS/MS analyses were performed on an Easy-nLC 1000 liquid chromatography system (Thermo Fisher Scientific) coupled to an Orbitrap Fusion Lumos via a nano-electrospray ion source (Thermo Fisher Scientific). Fractions from the first dimension RPLC were dissolved with loading buffer (5% methanol and 0.2% formic acid), and loaded onto a 360 μm I.D. × 2 cm, C18 trap column at a maximum pressure 280 bar with 12 μl solvent A (0.1% formic acid in water). Peptides were separated on 150 μm I.D. × 14 cm column (C18, 1.9 μm, 120 Å, Dr. Maisch GmbH) with a series of adjusted linear gradients according to the hydrophobicity of fractions with a flow rate of 600 nl/min. The MS analysis was performed in a data-dependent manner with full scans ($m/z$ 300–1400) acquired using an Orbitrap mass analyzer at a mass resolution of 120,000 at $m/z$ 200. The top speed data-dependent mode was selected for fragmentation in the HCD cell at normalized collision energy of 32%, and then fragment ions were transferred into the ion trap analyzer with the AGC target at 5e3 and maximum injection time at 35 ms. The dynamic exclusion of previously acquired precursor ions was enabled at 12 s.

**Label-free based MS quantification for proteins.** Raw MS files were processed with the MaxQuant software (version 1.5.3.30), using the integrated Andromeda search engine with FDR < 1% at peptide and protein level. As a forward database, the mouse refseq protein database (2013.07.01) was used. A reverse database for the decoy search was generated automatically in MaxQuant. Enzyme specificity was set to 'Trypsin', and a minimum number of seven amino acids were required for peptide identification. For label-free protein quantification, the 'Match Between Runs' option were used with a matching window of 3 min to transfer MS1 identification between Runs. For raw MS files from catTFRE sub-proteome, proteome and nuclear proteome, default settings were used for variable and fixed modifications (variable modification: acetylation (Protein-N terminus) and oxidation methionine (M), fixed modification: carbamidomethylation (C)). For raw MS files from phosphoproteome, setting for variable and fixed modifications included variable modifications for oxidation (methionine), acetylation (protein N-term), and phos-pho (STY) and fixed modifications for carbamidomethyl (C). For raw MS files from ubiquitylation proteome, setting for variable and fixed modifications included variable modifications for oxidation (methionine), acetylation (protein N-term), and GlyGly (K) and fixed modifications for carbamidomethyl (C). We used MaxQuant LFQ algorithm[46] to quantitate the MS signals, and the proteins' intensities were represented in iBAQ.

The iBAQ[30] (intensity-based absolute protein quantification) of each sample were transferred into FOT (a fraction of total protein iBAQ amount per experiment), and calculated z-score using the equation $z = (x-\mu)/\sigma$, ($\mu$ stands for the mean of the samples' FOT of one cycle, and $\sigma$ stands for the standard deviation of the samples' FOT of one cycle).

**Parallel reaction monitoring analysis for KC.** Data-dependent acquisition. Peptides were eluted from a 150 μm I.D. × 2 cm, C18 trap column and separated on a homemade 150 μm I.D. × 30 cm column (C18, 1.9 μm, 120 Å, Dr. Maisch GmbH) with a 150 min non-linear 5–35% ACN gradient at 600 nl/min. The combined method consisted of an MS1 scan at a resolution of 120,000 (at 200 $m/z$) with an AGC value of 3e6, max injection time of 80 ms and scan range from $m/z$ 300–1400, top30 precursor ions from MS1 were selected for MS2 scans with higher-energy collision dissociation detected in the Orbitrap first ($R = 15,000$ at 200 $m/z$, AGC target 2e4, max injection time 20 ms, isolation window 1.6 $m/z$, normalized collision energy of 27%, The dynamic exclusion of previously acquired precursor ions was enabled at 25 s).

Peptides of target GPs were identified and focused in the exact RT windows by data-dependent acquisition (DDA) scan (Supplementary Data 14). The parent ions in the table were monitored on an Easy-nLC system (Thermo Fisher Scientific, USA) coupled with Q-Exactive HF (Thermo Fisher Scientific, USA). Peptides were separated on a homemade 150 μm I.D. × 30 cm column (C18, 1.9 μm, 120 Å, Dr. Maisch GmbH) with a 150 min non-linear 5–35% ACN gradient at 600 nl/min.

The peptides were analyzed using full scan plus PRM modes. The full mass within the range of 300 to 1400 $m/z$ was collected. The MS1 resolution was set at 120,000 (at 200 $m/z$), MS2 methods were controlled with a timed inclusion list containing the target precursor $m/z$ value, charge, and a 3 min retention time window that was determined from DDA results. All of the raw files were processed using Skyline 3.1. The intensities of three fragment ions were summed for peptide quantification. The intensities of up to three peptides were summed and used for GP quantitative comparison.

**RNA-Seq.** About 0.1 g tissues were pooled together and extract total RNA using Trizol regent (Thermo Fisher Scientific). Total RNAs were subsequently enriched for polyadenylated RNA with the Oligotex mRNA Mini Kit (QIAGEN). For sequencing, we used Illumina HiSeqX platform with Pair End 150 reads, and sequence at depth averaging 50 million reads per sample. The results were aligned to mouse genome (UCSC version mm10) using tophat2 (v2.0.12), and transcript abundance was calculated using cuffnorm (v2.2.1), showed as FPKM (fragments per kilobase of exon model per million mapped reads) value.

The FPKM of each sample were calculated z-score using the equation $z = (x-\mu)/\sigma$, ($\mu$ stands for the mean of the samples' FPKM of one cycle, and $\sigma$ stands for the standard deviation of the samples' FPKM of one cycle).

**TF classification.** Proteins identified by DBA profiling were categorized into DBPs, TFs, and TCs. We extracted DBPs by filtering the genes' description "DNA-binding". And we extracted TFs and TCs by filtering the gene symbols, using the gene symbols list of TFs and TCs, from public databases described in previous studies[14].

**Bioinformatics and statistical analysis.** To determine the subset of cycling TFs, proteins, and transcripts. We performed JTK_CYCLE test with period range: 20–28 h and the amplitude and phase as free parameters. A statistical cut-off of $p < 0.1$ was used to define the cycling proteome, and $p < 0.05$ was used to define the cycling transcripts. Hierarchical clustering was performed using the pheatmap (Pretty Heatmaps) function in the R package (pheatmap, version 1.0.8). The GO terms that were enriched in the sets of enriched genes were determined using the Database for Annotation, Visualization and Integrated Discovery (DAVID) Bioinformatics Resource v 6.7 with Fisher's exact test.

**Bioinformatic analysis of kinase signaling pathway.** We extracted pathway-specific phosphorylated proteins and pathway-specific TFs by filtering their GO terms. The hierarchical network of kinase, phosphoproteome, and TFs were constructed based on multiple layers. The network among kinase and substrates were annotated using information accessed from PhosphoSitePlus[24], and the protein interaction network among TFs and phosphorylated proteins were generated with the STRING v10.0 using medium confidence (0.4) and experiments and database as active interaction sources. The network was visualized using Cytoscape v 3.3.0. The shortest path from pathway-specific phosphorylated proteins to TFs were based on the protein interaction data from STRING, and calculated using Rscript: shortest.paths.r.

**Bioinformatic analysis of Mediator-centered network.** We extracted pathway-specific TFs and TGs by filtering their GO terms. The hierarchical network of Mediators, TFs and TGs were constructed based on multiple layers. The protein interaction network among Mediators were performed with the STRING v10.0 using medium confidence (0.4) and "experiments and database" as active interaction sources. The network between Mediators and pathway-specific TFs were based on their Pearson's correlation coefficient, we linked each pathway-specific TF to three Mediators with highest Pearson's correlation coefficient with it. The network among TFs and TGs was obtained from CellNet[25], Data visualization was done with Cytoscape v 3.3.0.

**Bioinformatic analysis of KC and whole-liver protein network.** We extracted pathway-specific proteins by filtering their GO terms. The interaction between KC

**Fig. 7** The diurnal switch of the immune response in mouse liver. **a** Systematic workflow of the diurnal KC sub-proteome. **b** Diurnal oscillation of the number of KCs during the circadian cycle. **c** Bar plot shows distribution of the diurnal oscillated proteins of KCs at each ZT point. **d** Venn plot shows the overlap among diurnal rhythmic whole-liver proteins (JTK_CYCLE $p < 0.1$) and diurnal rhythmic KC proteins (JTK_CYCLE $p < 0.1$). **e** Scatterplot shows statistically enriched GO/KEGG pathways by diurnal rhythmic KC proteins and diurnal rhythmic whole-liver proteins. **f** Box plots shows the pathway-specific KC proteins' abundance of expression at ZT0 versus their expression at ZT12 (pair tailed Student's $t$ test $p < 0.05$). For the box plot, the bottom and top of the box are the first and third quartiles, and the band inside the box is the median of the proteins' abundance. **g** Schematic representation of diurnal changed TLR pathway-related proteins; for each protein, the corresponding expression level was represented by color. The color bar indicates normalized z-scored iBAQ. **h** Box plots show the stronger connection (more interactions) among immune-related KC proteins and whole-liver proteins in the daytime and stronger connection (more interactions) among lipid-related KC proteins and whole-liver proteins in the nighttime (pair tailed Student's $t$ test $p < 0.05$). For the box plot, the bottom and top of the box are the first and third quartiles, and the band inside the box is the median of the interactions between KC proteins and whole-liver proteins. **i** Diurnal interaction network among KC proteins and whole-liver proteins. KC upregulated proteins were colored in green, whole-liver upregulated proteins were colored in red. Diurnal rhythmic proteins (whole-liver proteins: JTK_CYCLE $p < 0.1$, KC proteins: JTK_CYCLE $p < 0.1$) were shown with red border, diurnal changed KC proteins (fold change > 5) were shown with black border

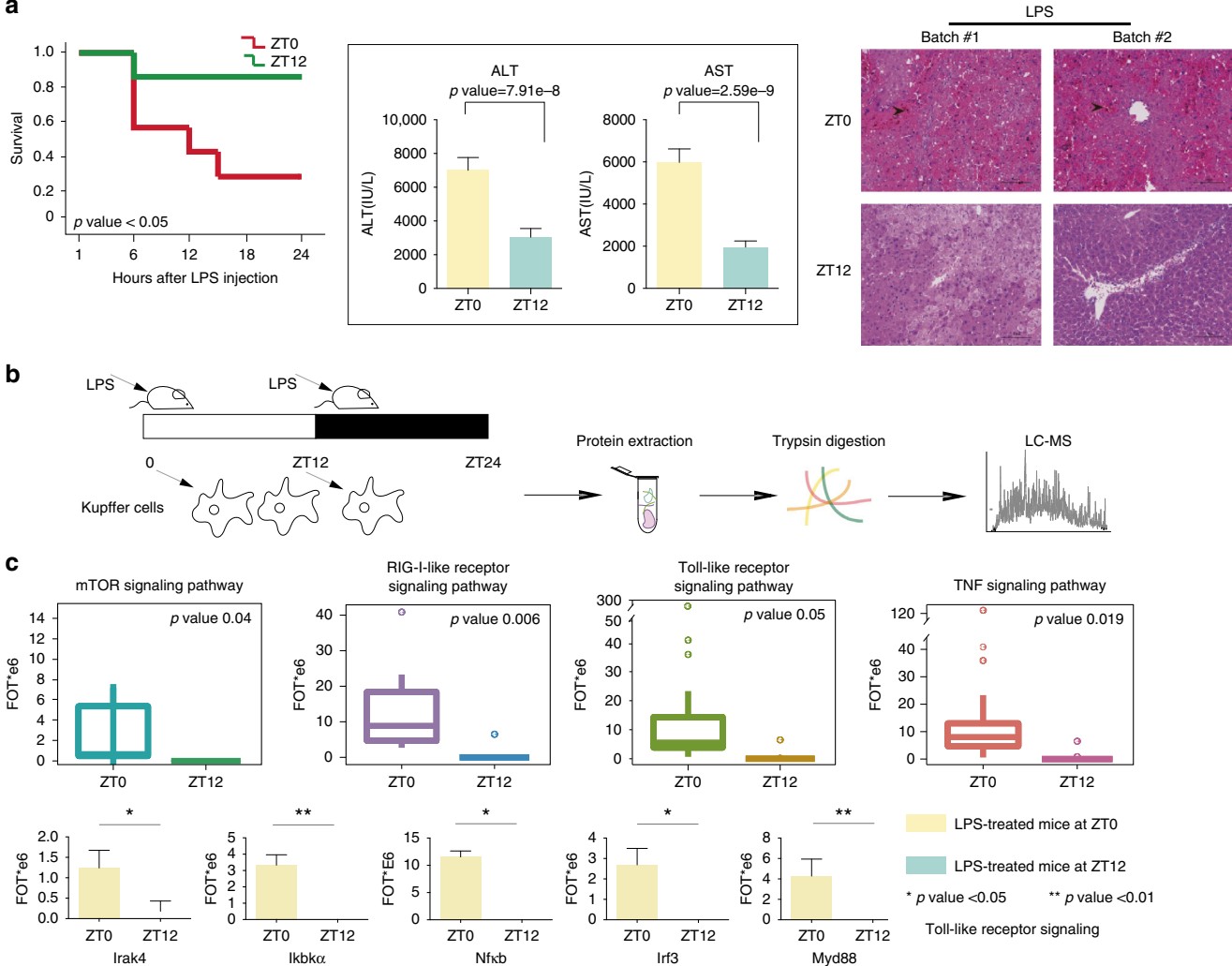

**Fig. 8** Liver's diurnal changed immune response to LPS. **a** Survival curves of mice after co-injected with LPS/D-GalN at ZT0 (red line) or ZT12 (green line) ($n = 6$ per group). Bar plots shows serum ALT, AST level of mice 6 h after been co-injected with LPS/D-GalN, at ZT0, or at ZT12. Data are mean SD ($n = 6$–8). Pair tailed Student's $t$ test was performed. Mice liver sections was prepared 6 h after LPS/D-GalN co-injection at ZT0 or at ZT12, stained with haematoxylin and eosin. The arrows indicated the focal necrosis area (scale bars = 100 μm). **b** Systematic workflow of the KCs' sub-proteome after LPS/D-GalN administration in circadian clock of mouse liver. **c** Box plots shows immune-related pathways were stronger activated by LPS/D-GalN at ZT0 than at ZT12. Bar plots shows higher expression of the TLR pathway-related proteins at 2 h after LPS/D-GalN administration at ZT0 or than at ZT12 (pair tailed Student's $t$ test $p < 0.05$). For the box plot, the bottom and top of the box are the first and third quartiles, and the band inside the box is the median of the proteins' abundance

proteins and whole-liver proteins were calculated based on information from protein–protein interaction database STRING (v10.0, medium confidence (0.4) and "experiments and database" as active interaction sources).

**Western blotting**. Denatured nuclear extractions or proteins were loaded onto polyacrylamide gels, and after electrophoresis, proteins were transferred onto PVDF membranes (Millipore), then blocked in 5% non-fat milk for 1 h at 25 °C. Incubation of primary antibodies was done overnight at 4 °C in 5% non-fat milk on TBS with 0.1% Tween-20. Primary antibodies were as follows: Med1 (CRSP1/TRAP220) (Bethyl, Cat.: A300-793A, 1:1000), Bmal1 (Cell Signaling Technology, Cat.: 14020 S, 1:1000), Rora (Proteintech, Cat.: 10616-1-AP, 1000), Pparδ (Proteintech, Cat.: 60193-1-Ig, Clone No.: 1B10E1, 1:1000), Tmem173 (Proteintech, Cat.: 19851-1-AP, 1:1000), Nfκb2 (Proteintech, Cat.: 10409-2-AP, 1:1000). The uncropped scans of blots of Bmal1, Rora, Pparδ, Med1, Tmem173, and Nfκb2 were represented in Supplementary Fig. 12.

**Data availability**. All Mass Spectrum raw data and the MaxQuant output tables have been deposited to iProX and can be accessed with the iProX accession: IPX0001145000, (TF DBA pattern and whole-liver proteome), IPX0001158000, (Nuclear proteome, phosphoproteome, ubiquitylation proteome and KC sub-proteome). RNA-seq data have been deposited to Sequence Read Archive (SRA), with accession number: SRP133633.

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

## Acknowledgements

This work is supported by National Key R&D Program of China (2017YFA0505102), National Natural Science Foundation of China (31770886, 31770892 and 31700682), Chinese Ministry of Science and Technology (2016YFA0502500), National Key R&D Program of China (2017YFC0908404), National Program on Key Basic Research Project (973 Program, 2014CBA02000), National Institute of Health (Illuminating Druggable Genome, Grant U01MH105026) and Grant for Shanghai Municipal Science and Technology Major Project (2017SHZDZX01), International Science & Technology Cooperation Program (2012DFB30080).

## Author contributions

C.D., J.Q., F.C.H. conceived the project and designed the experiment. Y.Z.W., L.S., M.W. L., R.Y.L., R.G., J.B.Q. conducted the experiments. Y.Z.W., LS, W.L.L, Q.Z. performed data analysis. Y.Z.W., C.D. wrote the manuscript. Y.W., J.Q., F.C.H., B.Z. edited the manuscript.

## Additional information

**Competing interests:** The authors declare no competing interests.

