## [Peer Review File · Nature Communications]

Reviewers' comments:

Reviewer #1 (Remarks to the Author):

In this manuscript, the authors carried out a multi-omics study to describe the circadian clock by establishing a transcription factor DNA-binding activity centered multi-omics landscape in mouse liver. The quantitative analysis were performed based on signaling transduction, nuclear protein expression, TF DNA-binding activity, mRNA expression, and the proteome profiling and protein ubiquitylation, revealing unique roles of the TF DNA-binding activity in circadian clock of the liver. Importantly, the authors found different regulatory omics-layers showed high diurnal consistence and cooperation in the four dominant rhythmic processes (immune response, glucose metabolism, fatty acid metabolism, and cell cycles).

The work is well designed, generating invaluable information to the research community for future studies. The paper will be of broad interest and is anticipated to have a broad impact on the fields of circadian rhythm, metabolism and liver biology.

This referee suggests some minor concerns before it can be published in Nature Communications.

Concerns

1. Many grammatical, spelling mistakes only in the manuscript. Some examples are listed below: line 52, 235, 395 (capital letter), 432, 447, 457 (past tense), 633 (ul and μ L), 634 (should be I.D.), 644, 685, 686, 688, 691, 711, 717, 721. The sentence 521-526 is too long. Some serious editing is necessary.

2. The reference styles are not consistent, for example, the journal name, line 746, 758, 828, 840, 784, 798 and 846; the info is not complete, line 813. In addition, the citations in manuscript are not consistent either.

3. Fig. 2C, the authors should describe the meaning of the upper bars.

In the bottom of Fig 3, for the p value, I can find $p < 0.1$ and $q < 0.1$, the author need to explain "q".

4. The title seems too long.

5. This story aims to reveal the transcription factor-centered multi-omics landscape. The authors reveal dynamic changes of many TFs in response to circadian rhythms, some of which changes DNA binding activity while protein abundance remains largely unchanged. It will be helpful to confirm such changes either using Western blot or immunostaining.

6. Suggest to make the story more concise; Remove some redundant statements in the paper.

Reviewer #2 (Remarks to the Author):

This manuscript by Wang et al. sets out on an ambitious goal of examining the signaling and regulatory networks in the context of the circadian clock in the liver. It is an impressive amount of work that employs a broad array of technologies. In general, the techniques that they have employed allow them to probe deeper than most other similar studies. They have found a number of molecules that other studies have missed. Nevertheless, although this manuscript has the potential to educate us about this topic more than any of the recent papers on this subject (Mauvoisin et al. PNAS, 2014; Wang et al. Cell Metabolism 2017; Mauvoisin et al. Cell Reports 2017), it really falls short. In addition, the manuscript is also hard to read, which makes it especially difficult. I feel that this is a borderline manuscript for Nature Communications, which would have to be improved significantly if it has to be published. I have the following issues with the manuscript:

Major issues:

1. They discuss the ubiquitylation data in passing. I believe that this can be very informative and they need to try to use this to better determine how this regulates the protein levels of proteins during the circadian cycle. This is a unique aspect of this study/
2. Did the authors find the changes observed in Kupfer cell analysis in the liver proteome data? Which proteins were found in this subset that were not found in the bulk liver proteomic data? Again, this is a unique aspect of this study.

Minor issues:

1. The authors state that "Four thousand and thirty-eight nuclear proteins were identified..." This is incorrect. What they mean to say is that they identified these proteins from their nuclear prep.
2. The authors should provide details of the RNA-seq experiments (e.g. what was the depth to which the samples were sequenced).
3. The authors should add a section on caveats of the label-free methods for protein quantitation used in the study.
4. The grammatical errors are too numerous to list. Following is a sampling (including many in the abstract itself):

Due to the technique limitations

signal transductions that rhythmically regulate the liver physiology remains undisclosed

Here, we presented TF DNA-binding activity

have evolved a system to adapt the physiological and anticipate diurnal variations

transcriptional factors

The expressions of enzymes

As its rapid development(16), the mass spectrometry(17)-based proteomics

Maria S. Robles et al. has quantified

we wonder if diurnal changed phosphorylated proteome

Inheritance from circadian transcriptome to proteome of mouse liver

We calculated the correlation coefficient

In consistence with the above

In order to investigated how

In figure 1 - "Ubiquitination" pattern

Response to Reviewers' comments:

Reviewers' comments:

Reviewer #1 (Remarks to the Author):

In this manuscript, the authors carried out a multi-omics study to describe the circadian clock by establishing a transcription factor DNA-binding activity centered multi-omics landscape in mouse liver. The quantitative analysis were performed based on signaling transduction, nuclear protein expression, TF DNA-binding activity, mRNA expression, and the proteome profiling and protein ubiquitylation, revealing unique roles of the TF DNA-binding activity in circadian clock of the liver. Importantly, the authors found different regulatory omics-layers showed high diurnal consistence and cooperation in the four dominant rhythmic processes (immune response, glucose metabolism, fatty acid metabolism, and cell cycles).

The work is well designed, generating invaluable information to the research community for future studies. The paper will be of broad interest and is anticipated to have a broad impact on the fields of circadian rhythm, metabolism and liver biology. This referee suggests some minor concerns before it can be published in Nature Communications.

Concerns

Q1: Many grammatical, spelling mistakes only in the manuscript. Some examples are listed below: line 52, 235, 395 (capital letter), 432, 447, 457 (past tense), 633 (ul and μ L), 634 (should be I.D.), 644, 685, 686, 688, 691, 711, 717, 721. The sentence 521-526 is too long. Some serious editing is necessary.

Reply: We corrected the mistakes in the revision.

Q2: The reference styles are not consistent, for example, the journal name, line 746, 758, 828, 840, 784, 798 and 846; the info is not complete, line 813. In addition, the citations in manuscript are not consistent either.

Reply: We revised the reference styles following the Nature Communications format and corrected the mistakes in the revision.

Q3: Fig. 2C, the authors should describe the meaning of the upper bars. In the bottom of Fig 3, for the p value, I can find $p < 0.1$ and $q < 0.1$, the author need to explain "q".

Reply: In Figure 2C, the upper white to black bar indicates the time series in 2 days' cycle. Daytime was shown in white, while nighttime was shown in black. To be clear, we added exact ZT points in the revised figures.

In the Figure 3, we combined our phosphoproteome data with previous published data by Roble et al. to build the kinase regulation network. The statistical test they used to detect rhythmic proteins are slightly different from ours: we identified rhythmic proteins using the JTK_CYCLE test with adjusted p value <0.1; while in their paper (*Cell Metabolism*, PMID: 27818261), the rhythmic proteins were detected using Perseus with q value (which means adjusted p value) <0.1. Both statistical algorithms were suitable for determining rhythmic proteins. We made this clear in the Figure legend in the revision.

Q4: The title seems too long.

Reply: In the revision, we changed the title to “A proteomics landscape of circadian clock in mouse liver”.

Q5: This story aims to reveal the transcription factor-centered multi-omics landscape. The authors reveal dynamic changes of many TFs in response to circadian rhythms, some of which changes DNA binding activity while protein abundance remains largely unchanged. It will be helpful to confirm such changes either using Western blot or immunostaining.

Reply: Thanks for the comments. To confirm our finding, we analyzed the expression of whole mouse liver proteins by performing PRM (Parallel Reaction Monitoring) and Western blotting in the revision. Unlike the rhythmic DNA-binding activity, protein expression levels of diurnal rhythmic TFs, such as Bmal1/Arntl, ppar δ , Ror α , and Stat5b, were not noticeably changed throughout the circadian cycle (**Figure CL1A, B**) (**Figure S3D, E in manuscript**).

Figure CL1 (A, B) Comparison between protein's abundance in the whole liver proteome and its DNA-binding activity. The temporal protein abundance of Bmal1, Rora, Ppar δ , conformed by Western Blotting (WB) in (A). X axis represents the sampled time points, Y axis represents ratio against protein's abundance of ZT0. The graph on the top presents the band blots of western blotting, with loading control. The temporal protein abundance of Pds5b, Stat5b, Ybx1, conformed by PRM-MS analysis in (B). X axis represents the sampled time points, Y axis represents ratio against protein's abundance of ZT0. Related to Figure S3.

Q6: Suggest to make the story more concise; Remove some redundant statements

Reply: We revised the manuscript and removed redundant statements as recommended.

Reviewer #2 (Remarks to the Author):

This manuscript by Wang et al. sets out on an ambitious goal of examining the signaling and regulatory networks in the context of the circadian clock in the liver. It is an impressive amount of work that employs a broad array of technologies. In general, the techniques that they have employed allow them to probe deeper than most other similar studies. They have found a number of molecules that other studies have missed. Nevertheless, although this manuscript has the potential to educate us about this topic more than any of the recent papers on this subject (Mauvoisin et al. PNAS, 2014; Wang et al. Cell Metabolism 2017; Mauvoisin et al. Cell Reports

2017), it really falls short. In addition, the manuscript is also hard to read, which makes it especially difficult. I feel that this is a borderline manuscript for Nature Communications, which would have to be improved significantly if it has to be published. I have the following issues with the manuscript:

Reply: We appreciate the insightful comments of the reviewer. In the revision, we comprehensively re-analyzed the ubiquitylation data of the liver in the circadian cycle. We thoroughly compared the KC proteome and whole liver proteome. Moreover, we have substantially re-written the manuscript to present the data more logically and the results more concisely.

The details are in the following point-to-point response:

Major issues:

Q1: They discuss the ubiquitylation data in passing. I believe that this can be very informative and they need to try to use this to better determine how this regulates the protein levels of proteins during the circadian cycle. This is a unique aspect of this study.

Reply: Thanks for the comments. Ubiquitylation is an important PTM in regulation, and its function in circadian rhythm is not well understood. We agree with the reviewer that in-depth analysis of these data, especially to find their connection with other datasets presented in this study would be very informative to reveal how circadian cycle is regulated, and this work also provides a rich resource to the research community. With this goal in mind, we re-analyzed the ubiquitylation data to illustrate its crucial role in circadian clock regulation.

We detected 3,424 ubiquitylation sites on 1,144 proteins. We found that the number of detected ubiquitylated proteins gradually decreased from the daytime to nighttime, while the whole liver proteome remained consistent around the circadian clock (**Figure CL2A**) (**Figure 5E in manuscript**). We grouped the ubiquitylated proteins according to the diurnal changes of their abundances and found that ubiquitylated proteins peaked in the daytime were enriched in the immune response and lipid metabolism, while the ubiquitylated proteins peaked in the nighttime were enriched in the glucose metabolism (**Figure CL2B**) (**Figure 5F in manuscript**).

We then analyzed the ubiquitylation pattern of TFs. We found that, in the ubiquitylation dataset, 5% of the rhythmic TFs including dominant diurnal rhythmic TFs such as Stat1 and Stat3, were detected as ubiquitylated. However, only 2% of the non-rhythmic TFs were detected as ubiquitylated. We also compared the ubiquitylation patterns of their target genes (TG). We found 8% TG of diurnal rhythmic TFs were ubiquitylated, while 6% TGs of non- diurnal rhythmic were ubiquitylated (**Figure CL2C**) (**Figure 5G**

in manuscript). Besides, as shown in Figure CL2D (**Figure 5H in manuscript**), the number of ubiquitylated TGs of the diurnal rhythmic TFs is significantly higher than the number of ubiquitylated TGs of the randomly selected TFs, indicating that target gene products were also further regulated by ubiquitylation, in addition to their regulation by TFs. The GO analysis had further revealed that the ubiquitylated diurnal rhythmic proteins significantly enriched in the major rhythmic pathways, such as glycolysis/gluconeogenesis, fatty acid degradation, PPAR signaling than the non-ubiquitylated rhythmic proteins (**Figure CL2E**) (**Figure 5I in manuscript**). To be more specific, we surveyed the ubiquitylation landscape of the innate immune response and found that the receptors (Il23r, Fcgr1g, Fcgr2b et al.), the enzymes and adaptors (Rac1, Trim25, Ikk, Traf1 et al.) and the effector TFs (Stat1 and Stat3) were all ubiquitylated. The same phenomenon was also found in the fatty acid metabolic process (**Figure CL2F**) (**Figure 5J in manuscript**).

Moreover, we also found consistency of the diurnal rhythmicity of proteins and their ubiquitylation level, for example, Trim25, Ikb, Irgm1 in the immune pathway, Me1, Slc27a5, and Apoa2 in the fatty acid metabolic pathway (**Figure CL2G**) (**Figure S9 in manuscript**). These results collectively suggest that ubiquitylation is another important mechanism in the regulation of diurnal rhythm in addition to transcription regulation

Figure CL2 (A) The number of ubiquitylated sites and proteins detected in our diurnal ubiquitylation

proteome. The Venn diagram shows the number of ubiquitylated proteins detected in each ZT point. Bar plot shows the comparison of numbers between ubiquitylated proteins with whole liver proteins detected at different ZT points. (B) Hierarchical clustering of the ubiquitylated proteins ordered by the phase of the oscillation. Values for each ubiquitylated protein at all analyzed samples (columns) are color code based on the intensities, low (blue) and high (yellow) z-scored normalized iBAQ. The upper white to black bar indicates the diurnal cycle. Daytime is shown in white, while night time is shown in black. (C) Bar plots on the left shows the percentage ratio between ubiquitylated diurnal rhythmic TFs and diurnal rhythmic TFs versus the percentage of ubiquitylated non-diurnal rhythmic TFs and non-diurnal rhythmic TFs. Bar plot on the right shows the percentage ratio between ubiquitylated diurnal rhythmic TFs' TGs and diurnal rhythmic TFs' TGs versus the percentage ratio between ubiquitylated non-diurnal rhythmic TFs' TGs and non-diurnal rhythmic TFs' TGs. (D) The number of ubiquitylated TGs of the diurnal rhythmic TFs versus the number of ubiquitylated TGs of the randomly selected TFs. (E) Scatterplot shows statistically enriched GO/KEGG pathways by ubiquitylated diurnal rhythmic proteins versus GO/KEGG pathways enriched by non-ubiquitylated diurnal rhythmic proteins. (F) Systematic overview of signal transduction participated by ubiquitylated proteins. Ubiquitylated proteins were colored based on their peak time. Ubiquitylated proteins peaked in the daytime were shown with red border, Ubiquitylated proteins peaked in the nighttime were shown with green border. (G) Temporal abundance of proteins and of their ubiquitylated forms. X axis represents the sampled time points, Y axis represents the proteins' abundance (normalized z-scored iBAQ). Relate to **Figure 5, Figure S9** in manuscript.

2. Did the authors find the changes observed in Kupffer cell analysis in the liver proteome data? Which proteins were found in this subset that were not found in the bulk liver proteomic data Again, this is a unique aspect of this study.

Reply: Thanks for the comments. According to the reviewer's suggestion, we deeply re-analyzed our cell-type resolved Kupffer (KC) proteome, and thoroughly compared cell-type resolved KC proteome with whole liver proteome.

Indeed, this is a unique aspect of the study. Among the 4,684 identified KC proteins, 690 proteins showed diurnal rhythm (p value < 0.1) (**Figure CL3 A**) (**Figure 7C in manuscript**). However, only 61 of them were observed in the whole liver proteome, suggesting that the cell-type resolved KC proteome provide more and unique information than the whole liver proteome (**Figure CL3 B**) (**Figure 7D in manuscript**). Moreover, we found the additional 1,078 proteins, although their abundance change didn't pass the JTK_CYCLE test, still showed strong oscillation between day and night (fold change > 5). These included many well-known immune related proteins such as Tmem173 and Mavs. We have confirmed many immune-related proteins with diurnal expression differences in the KC proteome but not in the whole liver proteome by MS and western blotting, such as Nfkb, Tmem173, Itgam, and so on (**Figure CL3 C, D**) (**Figure S11 in manuscript**).

We compared the GO enrichments of the diurnal changed KC proteins and whole liver proteins, and found the diurnal proteins in KC were more significantly enriched in the immune response pathways (**Figure CL3E**) (**Figure 7E in manuscript**).

We constructed an interaction networks between KC proteins and whole liver proteins around the circadian cycle. The connections between KC proteins and whole liver proteins in immune response pathway were more dominant in the daytime than the nighttime ($P = 1.5e68$), while the opposite was observed for the metabolism process, more connections between KC proteins and whole liver proteins were observed in the nighttime ($P = 1.5e18$) than in the daytime (**Figure CL3F, G**) (**Figure 7H, I in manuscript**). These results suggested a potential cooperation between KC and the liver organ.

Based on the above results, we employed the traditional TLR-induced liver injury models to investigate the differential immune response of the liver in the circadian clock, and profiled the KC proteome. The mice group treated with LPS/D-GalN in the daytime has poorer survival rate, higher ALT/AST (Alanine aminotransferase / Aspartate aminotransferase) level, and severer liver damage, compared to the group in the nighttime (**Figure 8A in manuscript**). Meanwhile, after the LPS treatment, the cell-type resolved KC proteome showed that the abundances of proteins of the innate immune pathway in KC were higher in the daytime, than in the nighttime, especially in the Tlr4 signal transduction pathway, which directly responds to LPS (**Figure 8C in manuscript**). These results indicated that the KC related immune response were elevated and may be more sensitive to LPS during the daytime.

A

B

C

D

E

F

G

Figure CL3 (A) Bar plot shows distribution of the diurnal oscillated proteins of KCs at each ZT point. (B) Venn plot shows the overlap among diurnal rhythmic whole liver proteins (p value<0.1) and diurnal rhythmic KC proteins (p value<0.1). (C) The temporal protein abundance of Nfkb2, Tmem173 in KCs, conformed by WB (Western Blotting). X axis represents the sampled time points, Y axis represents ratio against protein's abundance of ZT0. The graph on the top presents the band blots of western blotting, with loading control. (D) The temporal protein abundance of Arrb2, Itgam in KCs, conformed by DDA MS analysis. X axis represents the sampled time points, Y axis represents ratio against protein's abundance of ZT0. (E) Scatterplot shows statistically enriched GO/KEGG pathways by diurnal rhythmic KC proteins and diurnal rhythmic whole liver proteins. (F) Box plots show the stronger connection (more interactions) among immune related KC proteins and whole liver proteins in the daytime and stronger connection (more interactions) among lipid related KC proteins and whole liver proteins in the nighttime. (G) Diurnal interaction network among KC proteins and whole liver proteins. KC proteins were colored in green, whole liver proteins were colored in red. Diurnal rhythmic proteins (whole liver proteins: JTK_CYCLE p value<0.1, KC proteins: JTK_CYCLE p value<0.1) were shown with red border, diurnal changed KC proteins (fold change>5) were shown with black border. Related to **Figure 7** and **Figure S11**.

Q1: The authors state that “Four thousand and thirty-eight nuclear proteins were identified...” This is incorrect. What they mean to say is that they identified these proteins from their nuclear prep.

Reply: Thanks for the comments. In the revision, we changed our statement “Four thousand and thirty-eight nuclear proteins were identified in the two diurnal cycles in the mouse liver, including 105 TFs” into “We analyzed the liver nuclear proteome and identified 4,038 proteins from purified nuclei in the two diurnal cycles, including 105 TF”.

Q2: The authors should provide details of the RNA-seq experiments (e.g. what was the depth to which the samples were sequenced).

Reply: For RNA-seq experiment, we used Illumina HiSeqX platform, and sequence at the depth of averaging 50 million reads per sample. In the revision, we added detailed methods for our RNA-seq experiment in the Method section, with a subtitle “RNA-seq”.

Q3: The authors should add a section on caveats of the label-free methods for protein quantitation used in the study.

Reply: Compared with labeled based MS quantification, the label-free quantification can quantify unlimited number of samples for comparison and may also offer higher dynamic range to detect low abundance proteins (*J Biol Chem. PMID: 21632532*). Label-free quantification is widely applied in proteomic approaches, including the circadian rhythm studies (*Cell Metabolism, PMID: 27818260, Cell Metabolism, PMID: 27818261*). For this reason, we chose label free quantification as the method for our

diurnal proteome.

We revised the method section and added detailed methods for label-free based protein quantitation in the revision. with a title “**Label-free based MS quantification for proteins**”. We also added a section in the discussion. To be more specific, we used MaxQuant label-free quantification (LFQ) algorithm (*Nature*, PMID:21593866) to quantitate the MS signals, and the proteins’ intensities were represented in iBAQ (*Molecular & cellular proteomics*, PMID:24942700). The iBAQ (intensity based absolute protein quantification) of each sample were transferred into FOT (a fraction of total protein iBAQ amount per experiment), and calculated z-score using the equation $z=(x-\mu)/\sigma$, (μ stands for the mean of the samples’ FOT of one cycle, and σ stands for the standard deviation of the samples’ FOT of one cycle).

Q4: The grammatical errors are too numerous to list. Following is a sampling (including many in the abstract itself).

Due to the technique limitations

signal transductions that rhythmically regulate the liver physiology remains

undisclosed

Here, we presented TF DNA-binding activity

have evolved a system to adapt the physiological and anticipate diurnal

variations

transcriptional factors

The expressions of enzymes

As its rapid development(16), the mass spectrometry(17)-based proteomics

Maria S. Robles et al. has quantified

we wonder if diurnal changed phosphorylated proteome

Inheritance from circadian transcriptome to proteome of mouse liver

We calculated the correlation coefficient

In consistence with the above

In order to investigated how

In figure 1 – “Ubiquitinyion” pattern

Reply: The manuscript was now corrected by a native English writer.

REVIEWERS' COMMENTS:

Reviewer #1 (Remarks to the Author):

The authors have addressed my concerns. I suggest to publish as it is.

Reviewer #2 (Remarks to the Author):

The authors have now addressed all of my concerns and the manuscript is indeed substantially improved. I feel that it is now acceptable for publication.

Response to Reviewers' comments:

REVIEWERS' COMMENTS:

Reviewer #1 (Remarks to the Author):

The authors have addressed my concerns. I suggest to publish as it is.

Reviewer #2 (Remarks to the Author): The authors have now addressed all of my concerns and the manuscript is indeed substantially improved. I feel that it is now acceptable for publication.

Response: The referees did not raise any questions